# Redox regulation and dynamic control of brain-selective kinases BRSK1/2 in the AMPK family through cysteine-based mechanisms

George N Bendzunas[1†], Dominic P Byrne[2†], Safal Shrestha[3], Leonard A Daly[2,4], Sally O Oswald[2,4], Samiksha Katiyar[1], Aarya Venkat[1], Wayland Yeung[1,3], Claire E Eyers[2,4], Patrick A Eyers[3*], Natarajan Kannan[1,2*]

[1]Department of Biochemistry and Molecular Biology, University of Georgia, Athens, United States; [2]Department of Biochemistry, Cell and Systems Biology, Institute of Systems, Molecular and Integrative Biology, University of Liverpool, Liverpool, United Kingdom; [3]Institute of Bioinformatics, University of Georgia, Athens, United States; [4]Centre for Proteome Research, Institute of Systems, Molecular and Integrative Biology, University of Liverpool, Liverpool, United Kingdom

*For correspondence:
patrick.eyers@liverpool.ac.uk
(PAE);
nkannan@uga.edu (NK)

†These authors contributed
equally to this work

Competing interest: The authors
declare that no competing
interests exist.

Reviewing Editor: Amy H
Andreotti, Iowa State University,
United States

## eLife assessment

This study provides **fundamental** new knowledge into the role of reversible cysteine oxidation and reduction in protein kinase regulation. The data provide **convincing** evidence that intramolecular disulfide bonds serve a repressive regulatory role in the brain-selective kinases (BRSK) 1 and 2; part of the as-yet understudied 'dark kinome'. The findings will be of broad interest to biochemists, structural biologists, and those interested in the rational design and development of next-generation kinase inhibitors.

**Abstract** In eukaryotes, protein kinase signaling is regulated by a diverse array of post-translational modifications, including phosphorylation of Ser/Thr residues and oxidation of cysteine (Cys) residues. While regulation by activation segment phosphorylation of Ser/Thr residues is well understood, relatively little is known about how oxidation of cysteine residues modulate catalysis. In this study, we investigate redox regulation of the AMPK-related brain-selective kinases (BRSK) 1 and 2, and detail how broad catalytic activity is directly regulated through reversible oxidation and reduction of evolutionarily conserved Cys residues within the catalytic domain. We show that redox-dependent control of BRSKs is a dynamic and multilayered process involving oxidative modifications of several Cys residues, including the formation of intramolecular disulfide bonds involving a pair of Cys residues near the catalytic HRD motif and a highly conserved T-loop Cys with a BRSK-specific Cys within an unusual CPE motif at the end of the activation segment. Consistently, mutation of the CPE-Cys increases catalytic activity in vitro and drives phosphorylation of the BRSK substrate Tau in cells. Molecular modeling and molecular dynamics simulations indicate that oxidation of the CPE-Cys destabilizes a conserved salt bridge network critical for allosteric activation. The occurrence of spatially proximal Cys amino acids in diverse Ser/Thr protein kinase families suggests that disulfide-mediated control of catalytic activity may be a prevalent mechanism for regulation within the broader AMPK family.

## Introduction

Protein kinases are crucial components in cellular signaling networks, functioning as reversible molecular switches that orchestrate various biological processes. There are over 500 protein kinases encoded in the human genome that coordinate a wide range of cellular processes by catalyzing the transfer of a phosphate group from ATP to a hydroxyl group on the amino acid side chains of serine, threonine, or tyrosine residues in protein substrates (*Manning et al., 2002*). By catalyzing the reversible post-translational phosphorylation of Ser/Thr and Tyr residues of substrate proteins, protein kinases serve as signaling integrators that govern most aspects of eukaryotic life. Consequently, there exists a biological imperative to tightly control the catalytic activities of protein kinases, through cyclical phosphorylation of conserved amino acids, protein–protein interactions, and other regulatory post-translational modifications (PTMs). One essential mechanism governing kinase activity is the reversible phosphorylation of conserved amino acid residues within the activation loop, henceforth referred to as the T-loop (*Nolen et al., 2004*). In the inactive, unphosphorylated state, the T-loop adopts a wide range of conformations, including conformations that obstruct substrate binding (*Engh and Bossemeyer, 2001*). Phosphorylation of the activation loop induces an active spatial conformation that is typically more amenable to both binding and enzymatic phosphorylation of protein substrates, and this modification is prevalent across the kinase superfamily (*Faezov and Dunbrack, 2023*). Conversely, the removal of phosphate groups in this region by phosphatases (dephosphorylation) usually reverts kinases to an inactive state, generating a reversible switch to turn 'on' and 'off' kinase-dependent signaling pathways. More recently, we hypothesized that ~10% of the Ser/Thr human kinome may also be subject to a conserved form of redox-dependent regulation, including key members of the CAMK, AGC, and AGC-like families of kinases through reversible oxidation of an evolutionarily conserved Cys residue, which lies adjacent to the critical regulatory phosphorylation site on the activation loop (T-loop +2 position) (*Byrne et al., 2020*).

Understanding the molecular mechanisms underlying kinase regulation by redox-active Cys residues is fundamental as it appears to be widespread in signaling proteins (*Xiao et al., 2020*; *Corcoran and Cotter, 2013*; *Cao et al., 2023*) and provides new opportunities to develop specific covalent compounds for the targeted modulation of protein kinases (*Weisner et al., 2015*). Moreover, redox-active Cys are major sensors of reactive oxygen species (ROS), such as superoxide and peroxide, which function as endogenous secondary messengers to regulate various cellular processes (*Schieber and Chandel, 2014*; *Wani et al., 2011*). In particular, the high cell permeability of $H_2O_2$ relative to other ROS species allows it to be sensed intracellularly by reactive Cys, which can differentially impact protein function and cellular localization (*Lennicke et al., 2015*; *Rhee et al., 2005*). Chemically accessible and reactive Cys residues can transition through several redox states, such as the transient sulfenic acid species (Cys-SOH) and higher order, 'irreversible', sulfinic and sulfonic forms (Cys-SO$_2$H and Cys-SO$_3$H) (*Forman et al., 2017*; *Gupta and Carroll, 2014*). Importantly, in the context of allosteric protein redox regulation, the sulfenic oxidized Cys species can form disulfide linkages with other reactive Cys residues, while a sulfenic derivative has also been observed to be stabilized through the formation of a cyclic sulfenamide for tyrosine phosphatase PTP1B (*van Montfort et al., 2003*; *Salmeen et al., 2003*). The chemical reactivity, and thus biological susceptibility, of an individual Cys residue to oxidative modification is contingent on the intrinsic p$K_a$ value (where $K_a$ is the acid dissociation constant), which in turn is influenced by networks of interacting amino acids (including phosphorylated amino acids), solvent accessibility, protein–protein interactions, and protein structural dynamics (*Poole, 2015*; *Xiao et al., 2020*; *Soylu and Marino, 2016*). Unlike phosphorylation, which allosterically communicates with distal sites through positively charged residues that coordinate the phosphate group, it is largely unclear how the redox state of a T-loop localized Cys residue may alter the catalytic activity of a kinase (*Garrido Ruiz et al., 2022*), although a change in the activation segment conformation is a likely outcome, as demonstrated by careful analysis of Ser/Thr kinases, notably members of the AGC-family kinase AKT (*Su et al., 2019*).

The human AMPK-related kinase (ARK) family, consisting of 14 members (termed BRSK1-2, NUAK1-2, SIK1-3, MARK1-4, MELK, and AMPKα1 and AMPKα2), are fundamental regulators of cellular metabolism, growth, differentiation, and polarity (*Shao et al., 2014*; *Byrne et al., 2020*; *Shirwany and Zou, 2014*; *Zmijewski et al., 2010*), and BRSK1/2 function upstream of redox-based signaling to the pleiotropic transcription factor Nrf2 (*Tamir et al., 2020b*; *Tamir et al., 2020a*). Like other ARK members, BRSK1/2 possess similar structural organization, consisting of an N-terminal

serine/threonine catalytic (kinase) domain, which is followed by a ubiquitin-associated (UBA) domain, a C-terminal spacer, and in some members, a kinase-associated (KA1) domain (*Bright et al., 2009*; *Figure 1a*). In addition to sharing structural homology, all ARKs (except for MELK) are known to be activated by phosphorylation on their T-loop by the common upstream regulator LKB1, which is constitutively catalytically active in cells (*Lizcano et al., 2004*). All of the ARKs contain an activation loop 'T-loop +2 Cys' residue, which can be prognostic for redox regulation (*Byrne et al., 2020*), and the catalytic activities of several members have been demonstrated experimentally to be modulated by ROS, including the nominative member, AMPKα, which is both directly and indirectly regulated by redox-state (*Auciello et al., 2014*; *Hinchy et al., 2018*; *Choi et al., 2001*; *Shirwany and Zou, 2014*; *Shao et al., 2014*). However, the precise mechanisms whereby various ARKs are regulated under redox conditions remain obscure and are likely to be context specific.

The brain-specific Kinases (BRSKs, also termed Synapses of Amphids Defective [SAD] kinases), consist of two paralogs in vertebrates, termed BRSK1 and BRSK2, and are among the least well-studied of the ARK family (*Nie et al., 2012*). However, like all other members of the ARK family, BRSKs are downstream signaling targets of the Ser/Thr kinase LKB1 and also have the potential to be regulated 'upstream' by CAMKII, PAK1, and PKA, suggesting signal-dependent phosphorylation as a central regulatory mechanism (*Alessi et al., 2006*; *Lizcano et al., 2004*; *Nie et al., 2012*; *Bright et al., 2008*). BRSKs are highly expressed in the brain and central nervous system of model organisms, where they exhibit both distinct and redundant molecular functions (*Kishi et al., 2005*; *Wu et al., 2015*; *Nakanishi et al., 2019*); furthermore, they are implicated in several human pathologies, in particular neurodevelopmental disorders such as autism spectrum disorder (*Saiyin et al., 2017*; *Li et al., 2020*; *Deng et al., 2022*).

In the current study, we identify a new dominant mechanism for regulation of BRSKs through oxidative modification of conserved Cys residues within the kinase domain. We demonstrate that the catalytic activities of both BRSK1 and BRSK2 are fine-tuned through oxidative modification of the T-loop +2 Cys residue, which communicates with a BRSK-specific Cys residue in the APE motif (CPE in BRSKs) within the activation segment. We provide evidence that the T-loop Cys forms disulfide bonds with the 'CPE' motif Cys and that mutating the CPE-Cys to an alanine increases BRSK activity relative to the wild-type (WT) enzyme. Using a combination of biochemical analysis, structural modeling, and molecular dynamics (MD) simulations, we identify regulatory roles for these BRSK-conserved Cys residues and characterize novel intramolecular disulfide links, providing new insights into BRSK1/2 regulation and the broader AMPK family regulation. Together, these findings highlight complex regulatory processes for BRSK1/2 that are dependent on both phosphorylation and Cys-redox modulation, with broad implications for the other dozen members of the ARK family.

## Results

### Full-length BRSKs exhibit redox sensitivity

Full-length BRSK kinases share similar domain architecture to other ARK family members, including a UBA and kinase-associated domain (KA1) following their kinase domain (*Figure 1a*). Due to the absence of known endogenous substrates selectively phosphorylated by BRSK1 or 2 (*Tamir et al., 2020a*), we utilized a EGFP-Tau overexpression system in HEK-293T cells to assess BRSK activity (*Yoshida and Goedert, 2012*). BRSK1 and 2, when co-expressed with EGFP-Tau, induced substantial phosphorylation of Tau at Ser 262, a modification lost in kinase-dead (KD) mutants with the catalytic aspartate in the 'HRD' motif mutated to alanine (D146[BRSK1] or D141[BRSK2]), as shown in *Figure 1b*. The catalytic output of purified full-length human BRSK1 and 2 purified from Sf21 cells was next monitored in real time using a microfluidic kinase assays system and a generic ARK family substrate peptide AMARA (5-FAM- AMARAASAAALARRR -COOH), which is phosphorylated by BRSK1/2, but not the upstream kinase LKB1. In the absence of reducing agents (buffer alone), detectable peptide phosphorylation was extremely low for both kinases and ablated in the presence of $H_2O_2$ (*Figure 1c*). In contrast, inclusion of DTT enhanced BRSK1 and 2 activity by several orders of magnitude (*Figure 1c*). Moreover, $H_2O_2$-dependent inhibition of catalysis could be reversed, and even increased relative to basal activity, with the subsequent addition of a bolus of the reducing agent DTT (*Figure 1c*). BRSK proteins were rapidly activated by DTT in a concentration-dependent manner, suggesting an obligate requirement of an appropriate reducing environment in order to enable catalytic activity (*Figure 1d*

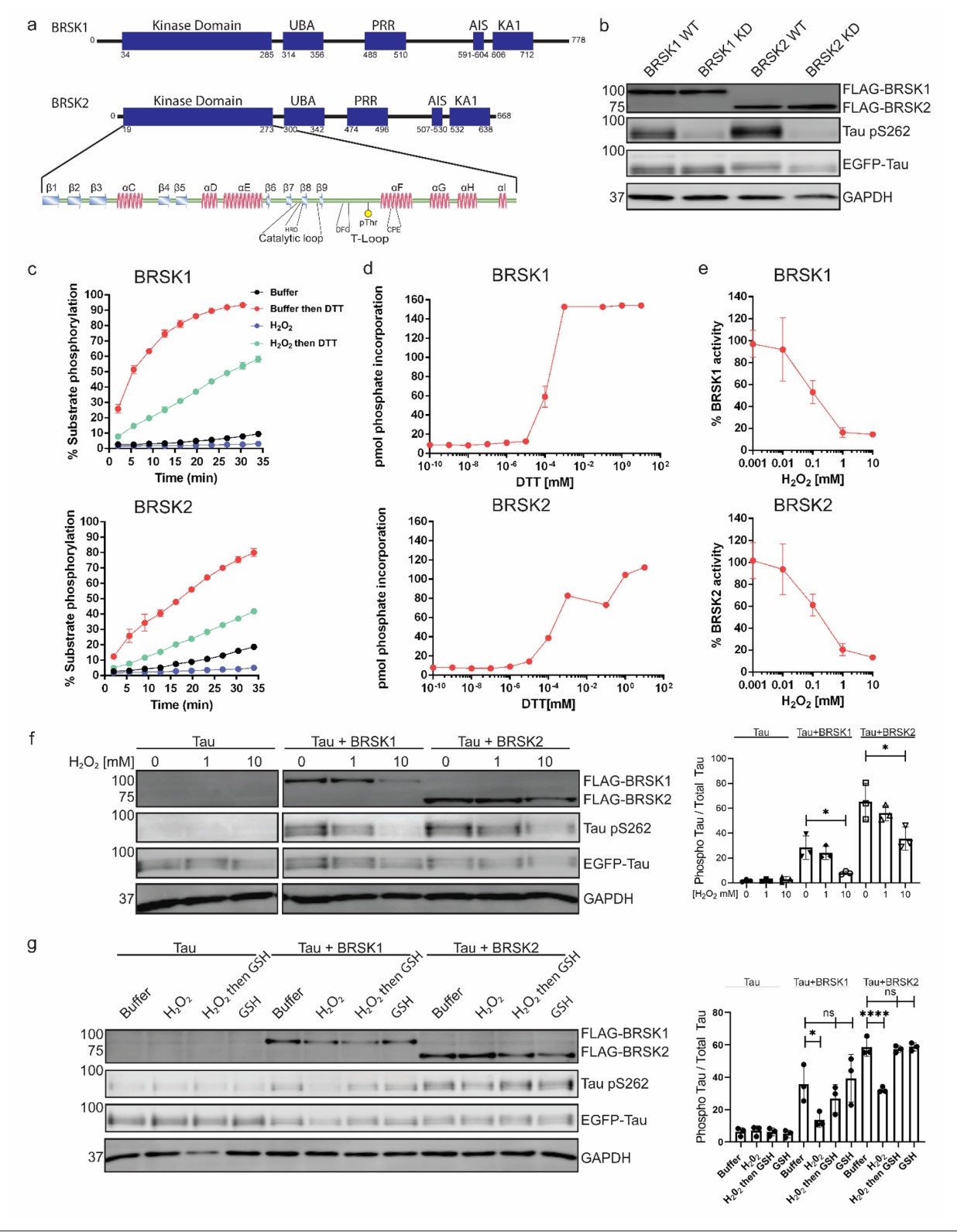

**Figure 1.** BRSK1/2 are redox sensitive. (**a**) Schematic representation of BRSK domain architecture, including kinase domain, ubiquitin-associated (UBA) domain, proline-rich region (PRR), kinase-associated domain (KA1), and autoinhibitory sequence (AIS). (**b**) Immunoblotting showing BRSK-dependent phosphorylation of Tau at Ser262 (pS262), from lysates of HEK-293T cells overexpressing full-length FLAG-BRSK1 or 2 (wild type [WT] or kinase dead [KD]) and EGFP-Tau. (**c**) Real-time phosphorylation of fluorescent AMARA peptide by full-length BRSK1 and 2 (200 ng) purified from Sf21 cells. BRSK

*Figure 1 continued on next page*

*Figure 1 continued*

proteins were incubated with buffer or 1 mM $H_2O_2$ for 10 min, reactions were then initiated with the addition of ATP and peptide substrate in the presence (where indicated) of 10 mM DTT. Dose–response curves for (**d**) DTT and (**e**) $H_2O_2$ with 200 ng full-length BRSK1 and BRSK2. All kinases assays are shown as mean and SD of three experiments. (**f**) Immunoblot (left) of pS262 in transiently co-transfected HEK-293T cells incubated with the indicated concentration of $H_2O_2$ for 10 min. Signal density for phospho Tau S262 and total Tau (GFP) was obtained using ImageStudio software (LI-COR) and results from at least three biological replicates were analyzed with GraphPad Prism software using one-way ANOVA to determine significance. Data shown is mean and SE. (**g**) Representative immunoblot (left) of transiently co-transfected HEK-293T cells treated with 10 mM $H_2O_2$ for 10 min before the addition of 20 mM GSH. Whole-cell lysates were harvested after a further 15 min. Normalized densitometry of Tau pS262 signal (right) was calculated from three independent experiments. Data shown is mean and SD. *$p<0.05$, **$p<0.01$, ***$p<0.001$.

The online version of this article includes the following source data and figure supplement(s) for figure 1:

**Source data 1.** Original western blots for *Figure 1*, indicating the relevant bands and treatments.

**Source data 2.** Original files for western blot analysis displayed in *Figure 1*.

**Figure supplement 1.** Redox regulation of BRSK1 and 2.

**Figure supplement 1—source data 1.** Original western blots for *Figure 1—figure supplement 1*, indicating the relevant bands and treatments.

**Figure supplement 1—source data 2.** Original files for western blot analysis displayed in *Figure 1—figure supplement 1*.

and *Figure 1—figure supplement 1a and b*). Similarly, basal BRSK activity was inhibited by a gradient of $H_2O_2$ (*Figure 1e*, *Figure 1—figure supplement 1c and d*). Western blotting revealed a dose-dependent and statistically significant decrease in BRSK-mediated pTau signal following incubation of HEK-293T cells with 10 mM peroxide for 10 min, with little alteration in total transfected Tau protein (*Figure 1f*, *Figure 1—figure supplement 1e and f*). At the highest concentrations of peroxide treatment, we detected a reduction in total BRSK protein levels, suggesting a potential loss of stability for both kinases. Chronic oxidative stress was next stimulated by supplementing culture medium with (2 U/ml) glucose oxidase to facilitate constitutive steady-state generation of $H_2O_2$ (*Askoxylakis et al., 2011*; *Mueller et al., 2009*; *Truong et al., 2016*). This revealed a time-dependent depletion of BRSK1 and 2-associated Tau phosphorylation (*Figure 1—figure supplement 1g and h*). Importantly, $H_2O_2$-dependent loss of pTau could be reversed following exposure of the cells to the physiological anti-oxidant glutathione (GSH) (*Figure 1g*). These findings suggest that reversible oxidative modulation is relevant to BRSK1/2 kinase-dependent signaling in human cells, which can be recapitulated in vitro.

## Mass spectrometric evidence that BRSK cysteine pairs can form intramolecular disulfide bonds

To identify residues that may contribute to redox regulation of BRSKs, we analyzed tryptic peptides derived from precipitated full-length cellular BRSK1 and BRSK2 by liquid chromatography–tandem mass spectrometry (LC-MS/MS). HEK-293T cells transiently over-expressing Strep-tagged BRSK proteins were lysed in the presence of the alkylating agent iodoacetamide to covalently block free thiol groups. LC-MS/MS revealed the presence of intramolecular bonds between C147[BRSK1]–C153[BRSK1] and C191[BRSK1]–C198[BRSK1] and C132[BRSK2]–C138[BRSK2] and C176[BRSK2]–C183[BRSK2] (*Figure 2a*). Of note, all identified disulfide forming Cys residues were located in the kinase domains of the two proteins, in close proximity to known catalytic or regulatory motifs. C147[BRSK1]–C153[BRSK1] and C132[BRSK2]–C138[BRSK2] structurally link the HRD motif in the catalytic loop to the preceding E-helix, and C191[BRSK1]–C198[BRSK1] and C176[BRSK2]–C183[BRSK2] couple the T-loop Cys to the Cys residue of the CPE motif in BRSK1/2 (equivalent to the APE motif in most kinase activation segments) (*Figure 2b*). To study these reactive Cys residues in the context of catalysis, we purified the unphosphorylated catalytic domain of human BRSK1[29-358] (BRSK1cat) or BRSK2[14-341] (BRSK2cat) to homogeneity from *Escherichia coli*. As expected, both truncated variants of BRSK were completely inactivated in our AMARA-based kinase assay, but could be 'switched on' following incubation with the physiological upstream regulator LKB1 (*Figure 2c*). Of note, despite sharing ~95% sequence identity within their kinase domain, LKB1-activated BRSK2 had higher catalytic activity compared to BRSK1 (*Figure 2c*). Moreover, and in support of our previous findings for full-length BRSK proteins (*Figure 1*), incubation of LKB1-activated WT BRSK1 or 2 with DTT greatly increased activity (*Figure 2c*). These data are consistent with regulatory Cys-based modification of the kinase domain under oxidative conditions, which can be reversed with a reducing agent in vitro.

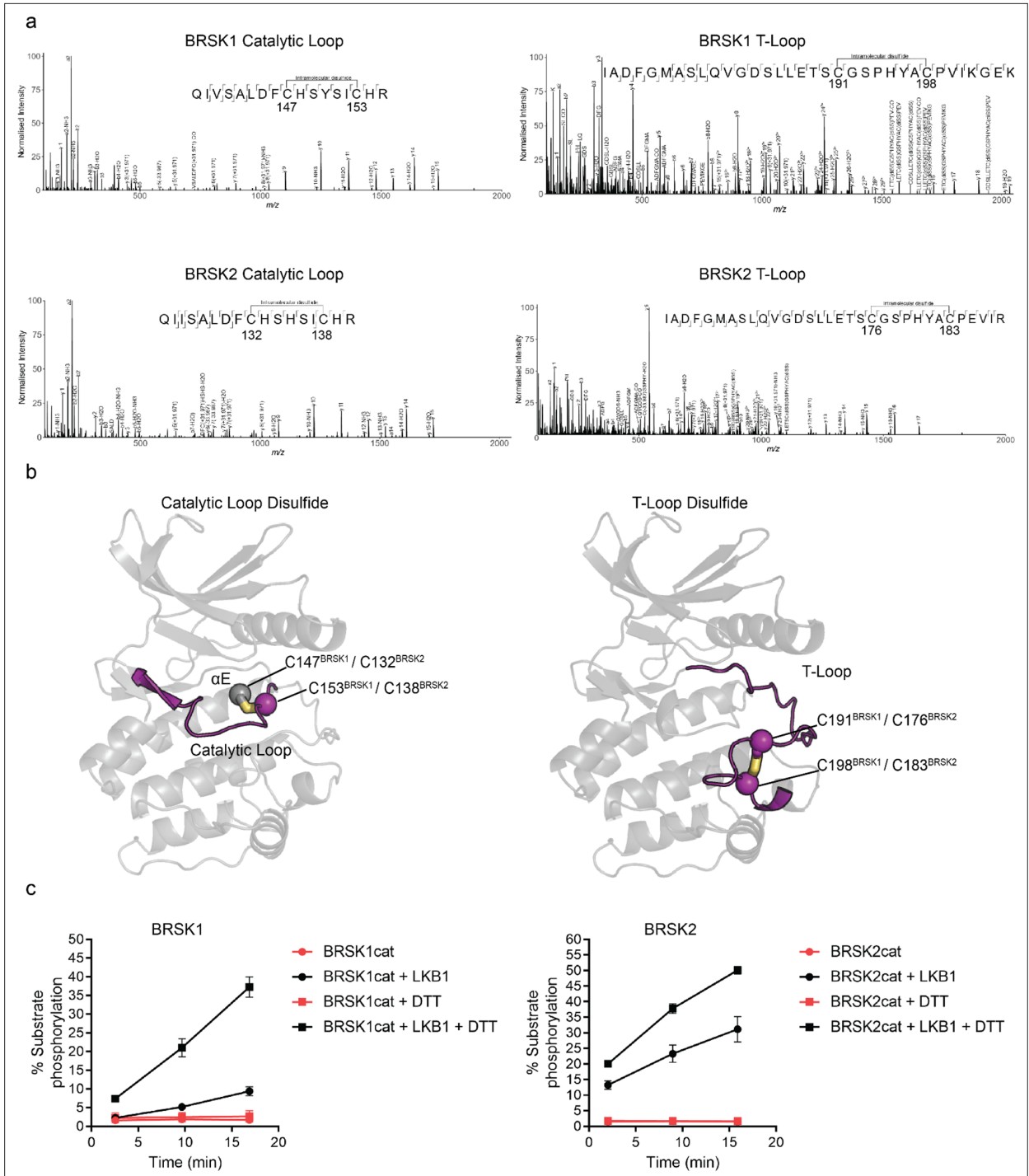

**Figure 2.** Intramolecular disulfide bonds form in the kinase domains of BRSK1 and 2. (**a**) Full-length BRSK1 and 2 were affinity-purified from HEK-293T cells and subjected to LC-MS/MS analysis. LC-MS/MS spectrum mapping revealed disulfide bridges formation between C147[BRSK1]–C153[BRSK1], C191[BRSK1]–C198[BRSK1], C132[BRSK2]–C138[BRSK2], and C176[BRSK2]–C183[BRSK2]. Peptide coverage was 74 and 78% for BRSK1 and BRSK2 respectively. (**b**) AlphaFold structures demonstrating the location of disulfide bonds within the kinase domains of BRSK1 and BRSK2. (**c**) Real time phosphorylation of fluorescent AMARA peptide by the kinase domains of BRSK1 and 2 (100 ng) purified from *E. coli*. BRSK1 (29–358) and BRSK2 (14–341) were activated by incubation with LKB1 and assayed in the presence or absence of 1 mM DTT.

## Emergence and structural location of cysteines residues in BRSK proteins

Reversible redox regulation of signaling proteins typically requires sulfenyl derivatization of an exposed Cys residue(s) (*Heppner et al., 2017*). Cys is the second least abundant amino acid in the vertebrate proteome, and conserved surface-exposed Cys side chains can function as redox 'hotspots' (*Fomenko et al., 2008*; *Su et al., 2019*; *Xiao et al., 2020*). Previously, we established that all 14 members of the ARK family kinases, including BRSK1 and 2, contain a T-loop +2 Cys residue. This residue is equivalent to the redox sensitive C199 found in PKA (*Humphries et al., 2002*) and is prognostic of redox regulation for multiple human Ser/Thr kinases (*Byrne et al., 2020*). Of the ARK family kinases that we previously analyzed, AMPKα1, SIK1-3, and MELK were all acutely inhibited by $H_2O_2$ in a reversible manner in vitro, which we attributed to sulfenylation of the activation segment Cys, based on biochemical and evolutionary analysis (*Byrne et al., 2020*). The T-loop +2 Cys correspond to $C191^{BRSK1}$ and $C176^{BRSK2}$ in BRSK1 and 2, respectively. This residue is located within the canonical activation segment, in close proximity to the regulatory site of LKB1 phosphorylation. Interestingly, mapping of Cys residues across the human ARK family reveals several conserved Cys located throughout their kinase domains (*Figure 3a and b*). However, these studies also reveal a distinguishing Cys residue that is unique to the catalytic domain of human BRSKs, which is located at the canonical alanine position of the 'APE' motif, converting it to "CPE" ($C198^{BRSK1}$/$C183^{BRSK2}$; *Figure 3b*). Of note, the unusual CPE Cys forms an intramolecular disulfide with the T-loop +2 Cys (*Figure 2a*). Intramolecular dimers incorporating T-loop Cys have also been identified in MELK and AKT2 (*Cao et al., 2013*; *Huang et al., 2003*). MELK is exceptional in that it possesses both a T-loop +1 as well as a T-loop +2 Cys, where the T-loop +1 Cys forms an intramolecular disulfide with a Cys proximal to the DFG motif and the T-loop +2 can form an intermolecular disulfide potentiating dimerization (*Cao et al., 2013*). In the case of AKT2, the T-loop +2 Cys forms an intramolecular disulfide with a Cys equivalent to that seen in MELK (*Huang et al., 2003*). In addition to the T +2 Cys, most human ARK family members (with the exception of MELK) contain an additional conserved Cys positioned seven residues upstream of the HRD motif (HRD −7 Cys) located in the E-helix (*Figure 3a and b*). BRSKs share the HRD −7 Cys ($C147^{BRSK1}$/$C132^{BRSK2}$), but further diverge from other ARK family members with the insertion of an additional potential disulfide bond-forming Cys residues preceding the HRD motif in the catalytic loop (CHRD-Cys, $C153^{BRSK1}$/$C138^{BRSK2}$ in *Figure 3a and b*).

## Phylogenetic analysis of BRSK protein sequences

A sequence-based analysis reveals the emergence of an early BRSK1 variant, which we term 'proto-BRSK1' that distinguishes it from the closely related AMPKs (*Figure 3—figure supplement 1a*). This is followed by a subsequent expansion of BRSK1 and 2 sequences that coincide with the appearance of vertebrates (*Figure 3c*). Sequence alignment of BRSK catalytic domains from a diverse array of organisms, including the ancestral paralog and invertebrate-specific proto-BRSK1, confirmed general sequence similarity and tight conservation of T-loop and HRD proximal Cys 'pairs' (*Figure 3d*). Interestingly, all BRSK domains also possess a Cys residue in the N-terminal β2-β3 loop ($C54^{BRSK1}$/$C42^{BRSK2}$), and BRSK2 contains an additional residue at this site, $C39^{BRSK2}$ (*Figure 3d*). The diversification of BRSKs from AMPKs also correlates with an increase in the total number of Cys residues in the kinase domain (*Figure 3—figure supplement 1*). Analysis of 2805 ARK-related sequences confirmed significant conservation of the T-loop +2 and HRD −7 Cys, which were found respectively in ~18% and ~10% of ePKs across diverse eukaryotic species (*Figure 3e*). These Cys residues were invariant in vertebrate BRSK sequences, as were the BRSK specific CPE and HRD −1 Cys residues (*Figure 3e*). Unsurprisingly, substitution of the APE Ala (PKA position 206, found in ~65% of ePKs) with a Cys is extremely uncommon (~1 %) in nearly all protein kinases, given the critical role of this motif in stabilizing the C-lobe and substrate interactions (*Figure 3—figure supplement 1b*). The distribution of amino acids at HRD −1 position is much more variable in ePKs, with Ile and Val being most commonly conserved (~36 and 30%, respectively) and a Cys appearing with similar low frequency (~2%; *Figure 3—figure supplement 1b*). The high degree of conservation observed for these Cys residues within vertebrate BRSKs indicates that they play critical functional or structural roles in these kinases (*Figure 3e*). This further suggests that diversification of the BRSKs in metazoans correlated with the accumulation of close proximity Cys 'pairs' with the potential to form regulatory disulfide bonds.

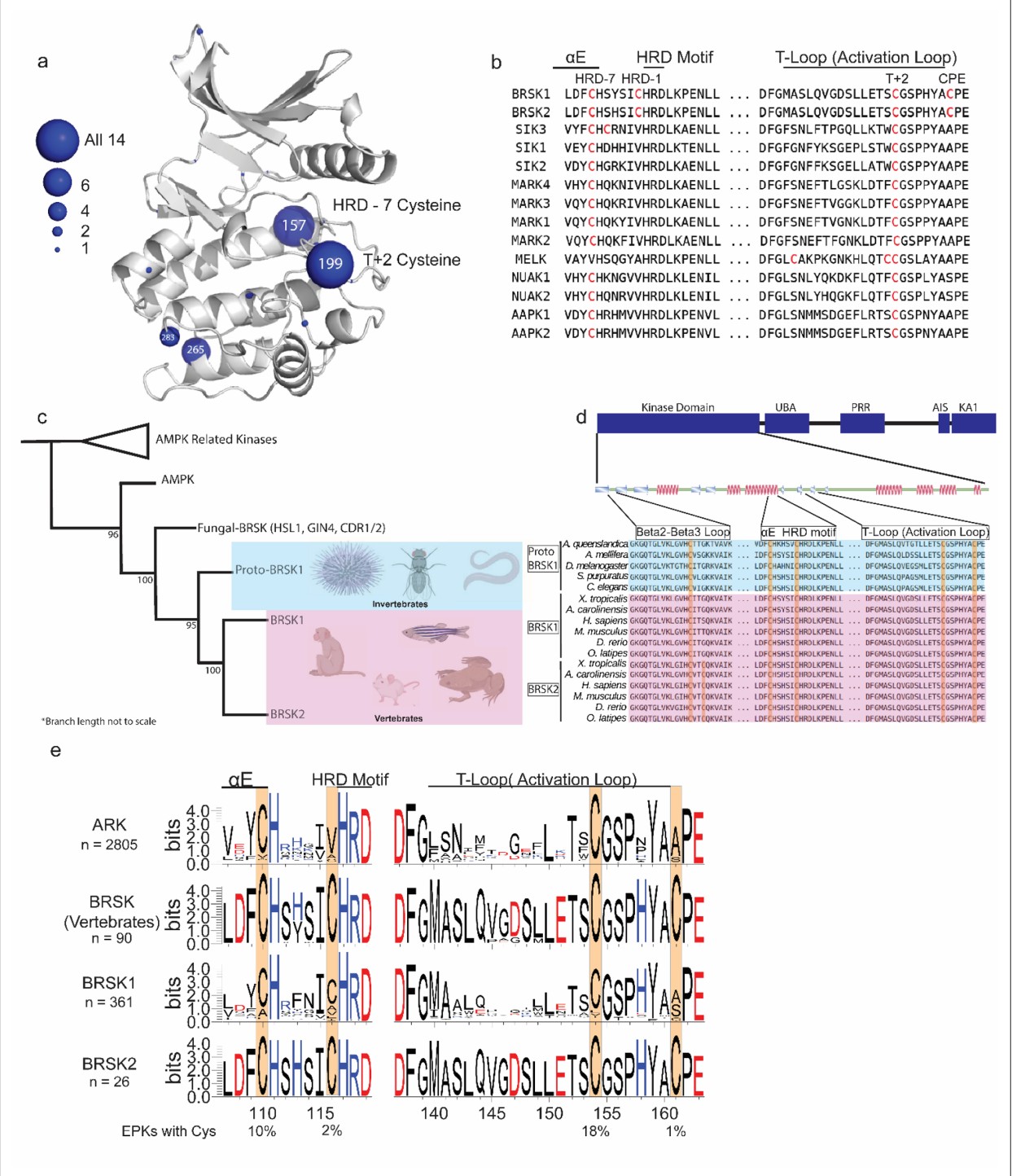

**Figure 3.** Cysteine pairs are highly conserved within the activation segments of BRSKs. (**a**) Mapping of Cys residues (spheres) in the kinase domains of human ARK family members. Numbers represent the corresponding amino acid position in PKA. Sphere size is proportional to the number of ARKs that contain a Cys at a specific site. (**b**) Activation segment sequence alignment of the 14 human ARKs. (**c**) Phylogenetic analysis showing divergence and grouping of BRSKs sub-families in different taxonomic groups. Bootstrap values are included for each clade. (**d**) Sequence alignment of the kinase domains of invertebrate and vertebrate BRSKs. (**e**) Analysis of relative amino acid conservation in ARKs and BRSKs, centered on the HRD containing catalytic loop, and the T-loop (between the DFG and APE motifs). Data is presented as hidden Markov models (HMM) Sequence Logos. The % of ePKs that possess a specific Cys is shown at the bottom.

The online version of this article includes the following figure supplement(s) for figure 3:

**Figure supplement 1.** Phylogenetic analysis of BRSK and the ARK family.

## Cysteine residues within the kinase domain fine-tune BRSK activity

To assess the role of BRSK domain Cys residues in modulating catalytic activity, we expressed and purified wild-type (WT) and Cys-to-Ala variants of the BRSK1 and 2 kinase domains in *E. coli* and performed in vitro kinase assays (*Figure 4*) The Cys-to-Ala variants included T-loop +2 Cys mutants (C191A[BRSK1] and C176A[BRSK2]), and T-loop CPE mutants (C198A[BRSK1] and C183A[BRSK2]), expressed either in a WT or mutant T-loop +2 Cys background (C191/198A[BRSK1] and C176/183A[BRSK2]). We also generated double mutants of the Cys residues upstream of the HRD motif (C147/153A[BRSK1] and BRSK2 C132/138A[BRSK2]), and the unique N-terminal Cys pair in BRSK2 (C39/42A[BRSK2]). All recombinant BRSK proteins were expressed in *E. coli* and purified without DTT. Crucially, we were able to detect intramolecular disulfide bonds (C191[BRSK1]- C198[BRSK1] and C176[BRSK2]- C183[BRSK2]) in the WT proteins by LC-MS/MS (*Figure 4—figure supplement 1*). Interestingly, we could only identify an HRD proximal disulfide bond (C147[BRSK1]–C153[BRSK1]) in BRSK1 under these specific experimental conditions (*Figure 4—figure supplement 1*). We next probed for mixed disulfide formation in the presence of glutathione using an antibody that recognizes glutathionylated proteins. We detected robust glutathionylation of both BRSK1 and BRSK2 in the presence of either reduced (GSH) or oxidized (GSSG) glutathione, and the signal strength inversely correlated with the presence of DTT (*Figure 4—figure supplement 2a*). Of note, all of the BRSK Cys-to-Ala mutants studied here could be readily glutathionylated, which supports the existence of multiple reactive Cys residues within the kinase domains of BRSK1 and 2. To detect alterations in redox regulation, all BRSK proteins were first activated by incubation with LKB1, and T-loop phosphorylation was confirmed by immunoblotting (*Figure 4—figure supplement 2b*). The active BRSK1/2 kinases were then assayed using the specific AMARA peptide in the presence or absence of fixed concentrations of DTT, and kinase activity was normalized to pBRSK signal derived from *Figure 4—figure supplement 2b*. In agreement with our previous findings with full-length BRSKs, DTT was strongly activating for WT variants of both kinases, and this effect was severely blunted for the T-loop +2 Cys-Ala mutants, which exhibited lower basal rates of peptide phosphorylation. This is entirely consistent with our previous observations for Cys-based mutants of analogous residues in a range of distinct Ser/Thr kinases (*Byrne et al., 2020*; *Figure 4a and b*). Of note, despite sharing ~95% sequence identity within their kinase domain, LKB1-activated BRSK2 demonstrated enhanced catalytic activity compared to BRSK1 (*Figure 4b* compared to *Figure 4b*). Perhaps unsurprisingly, given their distant location on an N-lobe loop, mutation of the BRSK2 exclusive C39[BRSK2] and C42[BRSK2] residues had limited effect on the activity of BRSK2 (*Figure 4b*). In contrast, tandem mutation of the HRD proximal Cys residues resulted in pronounced abrogation of kinase activity, regardless of assay conditions for both kinases (*Figure 4a and b*). Given the near absolute conservation of the HRD –7 Cys in the ARK family of protein kinases, it is possible that this residue (C147[BRSK1] and C132[BRSK2]) is functionally important for catalytic activity in some yet unidentified capacity. Interestingly, mutation of the CPE motif Cys (T-loop +9 Cys), and de facto restoration of the canonical APE motif, were insufficient to blunt DTT-dependent activation of either kinase. Moreover, this mutation, which would eliminate C191[BRSK1]–C198[BRSK1] and C176[BRSK2]–C183[BRSK2] disulfide bonds, increased basal (non-DTT stimulated) catalytic activity by 1.5–2-fold for both kinases. WT forms of BRSK2, and in particular BRSK1, were strongly inhibited by oxidative conditions, even when assays were preceded by DTT-dependent activation (*Figure 4c and d*). Unsurprisingly, the low levels of detectable C191A[BRSK1] and C176A[BRSK2] activity that could be measured following stimulation by DTT were completely abolished following the addition of H$_2$O$_2$. In contrast, CPE mutants (C198A[BRSK1] and C183A[BRSK2]) were sharply activated by DTT but still exhibited further oxidative inhibition (*Figure 4c and d*), although to a lesser extent than their WT counterparts, particularly in the case of BRSK1.

To ensure that the observed variations in activity between variants of BRSK1/2 were not a consequence of structural impairment, we also performed differential scanning fluorometry (DSF) to assess protein folding and stability. Incubation of WT BRSK1 and 2 with DTT had no measurable effect on the thermal stability of either protein, suggesting that chemical disruption of pre-formed disulfide bonds had a minimal detectable impact on global protein stability, despite greatly increasing kinase activity (*Figure 4—figure supplement 2c*). These assays also revealed only minor perturbations in protein thermal stability due to the incorporation of specified Cys-to-Ala mutants. Interestingly, we observed a consistent decrease in $T_m$ values for C147/153A[BRSK1] and C132/138A[BRSK2] ($\Delta T_m$ ~ –2°C), suggesting a modest decrease in protein stability, and increased $T_m$ values for CPE mutants (C198A[BRSK1] and C183A[BRSK2]; $\Delta T_m$ ~+3°C) (*Figure 4—figure supplement 2d*).

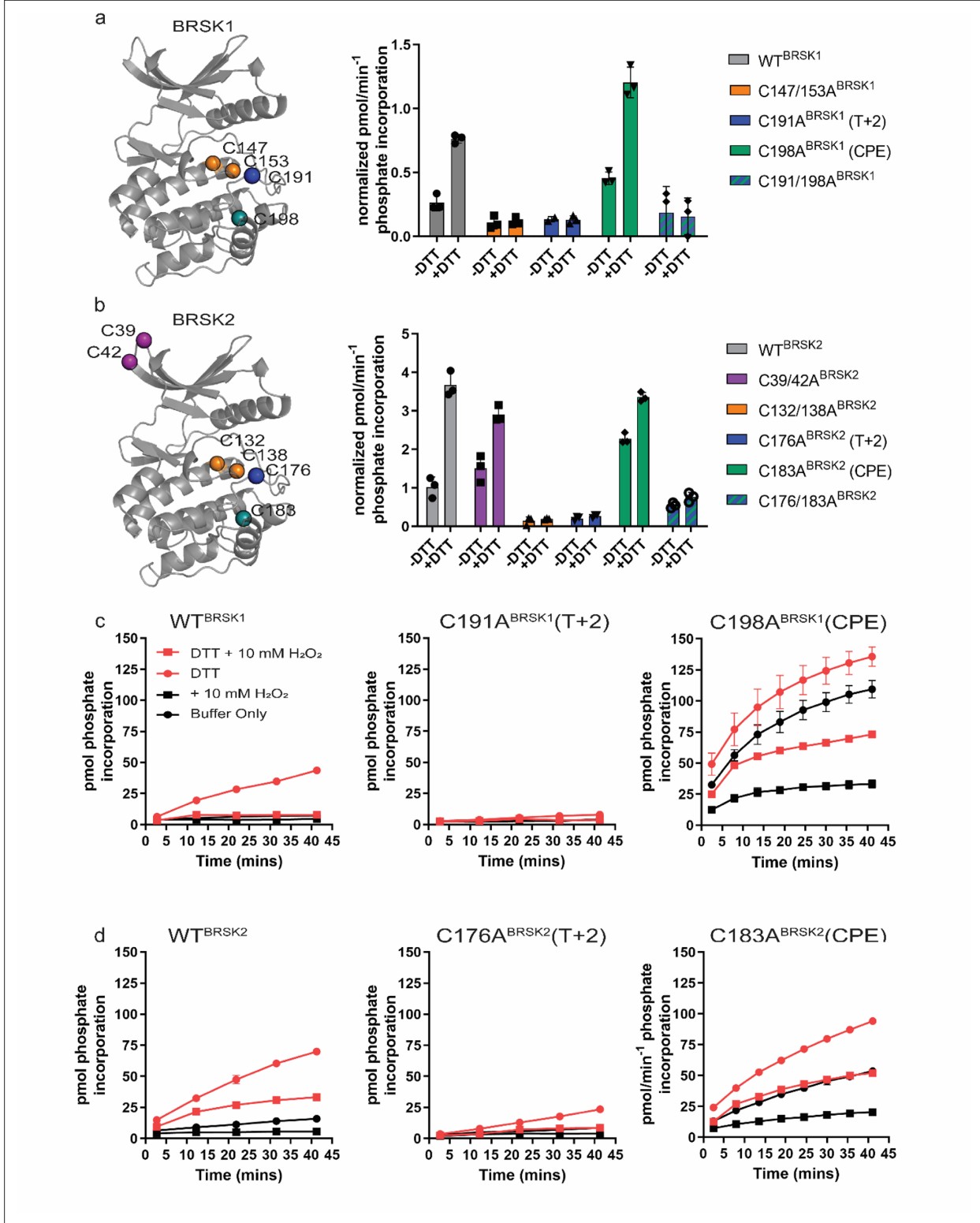

**Figure 4.** Cysteine residues within the kinase domain fine-tune BRSK activity. In vitro kinase assays (right panels) showing normalized rates of peptide phosphorylation by WT and Cys-to-Ala variants of (**a**) BRSK1 and (**b**) BRSK2. 100 ng of LKB1-activated BRSK kinase domain was assayed in the presence or absence of 1 mM DTT. The positions of mutated Cys residues are modeled on the kinase domain as colored spheres (left panel). Real-time in vitro assays using (**c**) 50 ng BRSK1 and (**d**) 20 ng BRSK2. LKB1-activated BRSK proteins were incubated on ice in the presence or absence of 250 μM DTT for 30 min. Assays were initiated by the addition of ATP and fluorescent peptide substrate in the presence or absence of 1 mM $H_2O_2$. All data are mean and SD of three experiments and activities are normalized to LKB1-phosphorylated BRSK signal.

*Figure 4 continued on next page*

*Figure 4 continued*

The online version of this article includes the following source data and figure supplement(s) for figure 4:

**Figure supplement 1.** LC-MS/MS analysis of BRSK1/2 catalytic domains.

**Figure supplement 2.** Biochemical analysis of BRSK Cys-to Ala mutants.

**Figure supplement 2—source data 1.** Original western blots for *Figure 4—figure supplement 2*, indicating the relevant bands and treatments.

**Figure supplement 2—source data 2.** Original files for western blot analysis displayed in *Figure 4—figure supplement 2*.

## Cellular analysis of BRSK Cys-based regulation

We next evaluated the relative contributions of the conserved T-loop Cys residues to BRSK redox sensitivity in a cellular context using our EGFP-Tau HEK-293T co-expression system and full-length BRSK proteins. Mirroring our peptide-based kinase assays, loss of the T-loop +2 Cys residue evoked a marked decrease in BRSK-dependent Tau phosphorylation (*Figure 5a and b*). In contrast, mutation of the CPE Cys to an alanine consistently increased overall Tau phosphorylation (~1.5- and ~1.2-fold increase relative to WT BRSK1 and BRSK2, respectively) (*Figure 5*). Interestingly, the CPE mutations preserved BRSK redox sensitivity in cells treated with hydrogen peroxide, and inclusion of GSH was sufficient to restore BRSK-dependent pTau signals. Finally, we extended our analysis to consider the BRSK1 and 2 HRD motif proximal cysteines, and the BRSK2 exclusive C39/C42 pair. As predicted, Tau phosphorylation by BRSK2 C39/42A (which closely matched the activity profile of WT BRSK2 in our in vitro kinase assays; *Figure 4*) was comparable to that observed for WT (but still less than

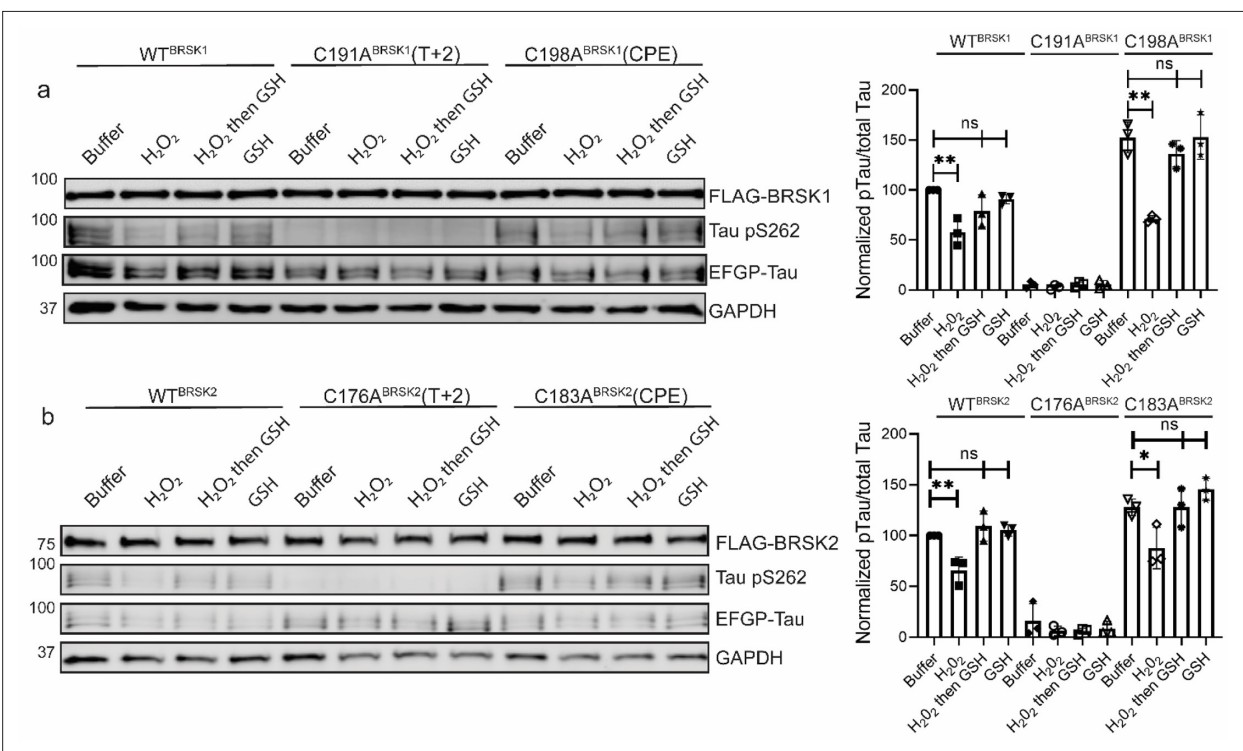

**Figure 5.** Impact of T-Lloop and CPE Cys-to-Ala mutations on BRSK redox sensitivity in a cellular EGFP-Tau HEK-293T co-expression system. Representative immunoblot of EGFP-Tau co-expressed with WT and Cys-to-Ala mutants of (**a**) BRSK1 and (**b**) BRSK2 (left panels). Transiently transfected HEK-293T cells were treated with or without 10 mM $H_2O_2$ for 10 min before the addition of 20 mM GSH. Whole-cell lysates were harvested after a further 15 min. Signal density for phospho Tau S262 and total Tau (GFP) was obtained using ImageStudio software (LI-COR) and results from at least three biological replicates were analyzed with GraphPad Prism software using one-way ANOVA to determine significance (right panels). Data shown is mean and SE. All values are normalized to Tau pS262 signals from control (buffer only treatment) WT BRSK and Tau co-transfections. Data shown is mean and SD. *$p<0.05$, **$p<0.01$, ***$p<0.001$.

The online version of this article includes the following source data for figure 5:

**Source data 1.** Original western blots for *Figure 5*, indicating the relevant bands and treatments.

**Source data 2.** Original files for western blot analysis displayed in *Figure 5*.

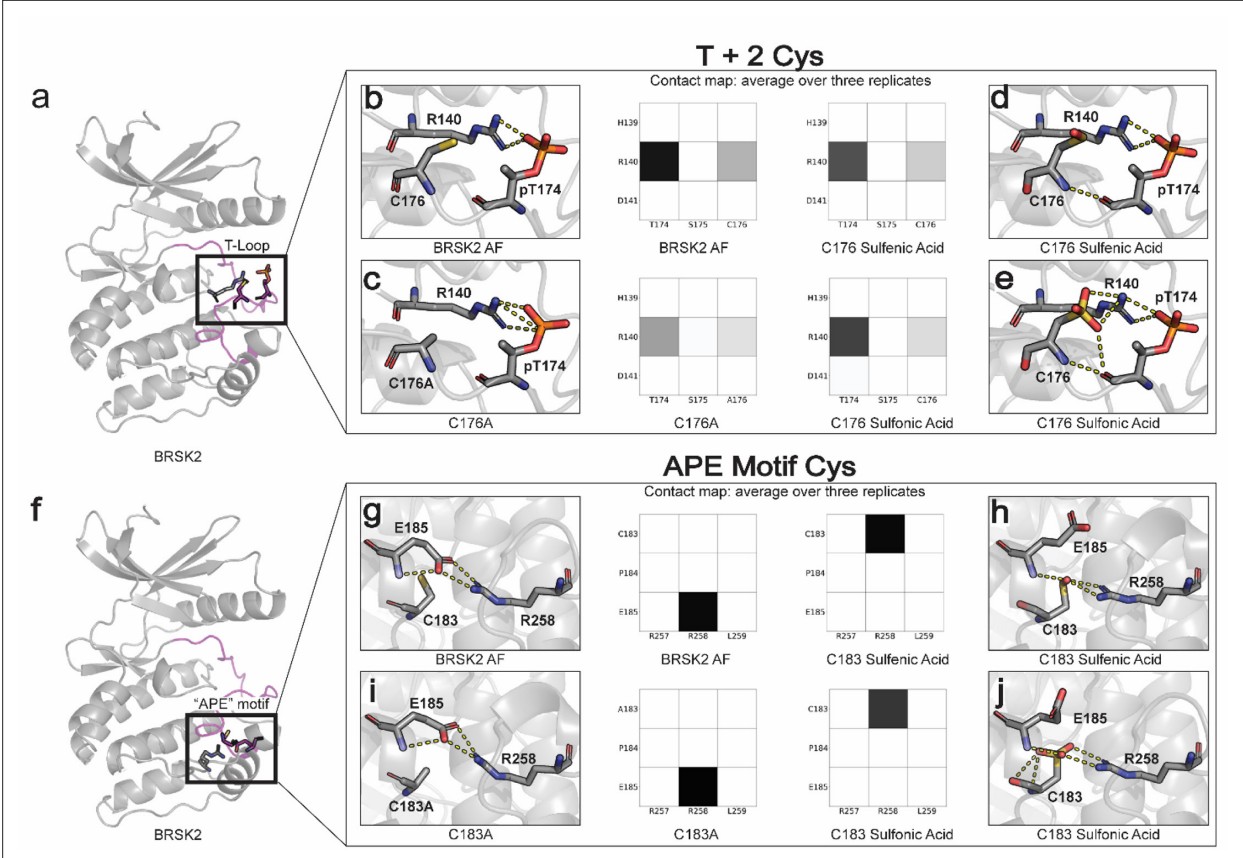

**Figure 6.** Oxidative cysteine modifications alter critical structural interactions required for BRSK allosteric regulation. Three replicates of 100 ns GROMACS molecular dynamics simulations were performed to evaluate the effects of cysteine mutation and oxidation. Salt bridge disruption was analyzed by generating contact maps representing the percentage of the simulation time in which residues were within appropriate distance (3 Å). (**a**) T +2 Cys is located in proximity to the activation loop threonine in the T loop. (**b–e**) Evaluation of pT174-R140 salt bridge formation in wild-type, C176A, and oxidized C176 BRSK2. (**f**) Location of CPE salt bridge within BRSK2. (**g–j**) Evaluation of E185-R258 salt bridge formation in wild-type, C183A, and oxidized C183 BRSK2.

The online version of this article includes the following figure supplement(s) for figure 6:

**Figure supplement 1.** Molecular dynamics simulations of intramolecular disulfide bonds.

hyper-active BRSK2 C183A) and was also similarly inhibited by the presence of $H_2O_2$ (*Figure 4—figure supplement 2e*). Using the AMARA peptide as a substrate, we previously demonstrated that BRSK1 C147/153A and BRSK2 C132/138A were catalytically compromised (in a manner resembling the respective T-loop +2 Cys-Ala mutants; *Figure 4*). It is consistent that BRSK2 C132/138A was unable to increase pTau signal above background levels (*Figure 4—figure supplement 2e*). Finally, we were unable to detect BRSK1 C147/153A protein expression in transfected cell lysates, which may indicate a loss of stability for this protein (*Figure 4—figure supplement 2e*).

## Cysteine modifications alter critical structural interactions required for kinase allosteric regulation

We next sought to investigate the structural basis for redox-dependent regulation of BRSK activity using molecular modeling and MD simulations. Our in vitro analysis established that oxidative conditions inhibit the active, T-loop phosphorylated form of BRSKs, and so our simulations were performed on an 'active' conformation of BRSK2 generated using AlphaFold2 (see 'Materials and methods'). Cysteine residues can undergo both reversible (sulfenic) and irreversible (sulfonic) oxidation, and so sulfenic acid or sulfonic acid forms of Cys were incorporated at the $C176^{BRSK2}$ and $C183^{BRSK2}$ positions. Additionally, the impact of a non-redox-active 'silent' Ala residue was also modeled at these sites.

The T +2 C176[BRSK2] is in close proximity to threonine T174[BRSK2], phosphorylation of which stabilizes the kinase domain in an active conformation through salt bridge interactions with charged residues in the catalytic loop (*Figure 6a*). In particular, R140[BRSK2] in the canonical HRD motif coordinates with the phosphate group of pT174[BRSK2] (*Nolen et al., 2004*). Simulations demonstrate that the R140[BRSK2]-pT174[BRSK2] salt bridge is preserved across the entire MD simulation, as demonstrated by the contact map (*Figure 6b*). In the C176Ala[BRSK2] simulations, the coordination between R140[BRSK2] and pT174[BRSK2] is partially attenuated due to an increase in the flexibility of pT174[BRSK2] (*Figure 6c*). This predicted increase in flexibility may explain the loss of BRSK2 catalytic activity for C176A[BRSK2] mutant (*Figure 4*). However, oxidative modification of C176[BRSK2] did not result in a substantial disruption of the salt bridge interaction (*Figure 6d and e*). As such, it is unclear at this stage precisely how oxidation of the T +2 Cys exerts its regulatory effect on BRSK2 kinase activity.

In contrast to C176[BRSK2], C183[BRSK2] within the CPE motif is buried in the C-terminal lobe of the kinase domain, and the SH group of C183[BRSK2] is pointed toward a canonical salt bridge that forms between the glutamate (E185[BRSK2]) in the APE/CPE motif and R259[BRSK2] in the I-helix (*Figure 6f*). The E185-R259 salt bridge is a eukaryotic protein kinase (EPK)-specific interaction that is critical for maintaining the EPK fold and for allosterically coupling the T-Loop to distal substrate binding and regulatory sites (*Nie et al., 2012*; *Oruganty and Kannan, 2012*). The selective conservation of Cys in place of Ala in the APE motif represents an interesting divergence of BRSKs from other ARK family kinases (*Figure 3e*, *Figure 3—figure supplement 1*). When C183[BRSK2] is in a reduced form or mutated to an alanine, the E185-R259 is maintained throughout the MD simulation (*Figure 6g and i*) Remarkably, in simulations incorporating oxidative modification of C183[BRSK2] we observed the immediate breaking of the E185-R259 salt bridge, and this contact remains broken throughout the simulation (*Figure 6h and j*). Oxidation of C183[BRSK2] to either sulfenic or sulfonic acid rewires this salt bridge, with R259[BRSK2] exclusively interacting with the oxidized C183[BRSK2] while E185[BRSK2] pivots outward and becomes more solvent-exposed. Thus, oxidized C183[BRSK2]-mediated disruption of E185-R259 [BRSK2] salt bridge represents a unique inactive state in BRSKs which breaks the allosteric network that allows cross-communication between the T-loop and the C-lobe.

Surprisingly, simulations incorporating intramolecular disulfide bonds identified in MS/MS experiments did not indicate any major changes in dynamics resulting from either the Cys132-138 or the Cys176-183 disulfide bond formation. Most of the fluctuations in these simulations were confined to

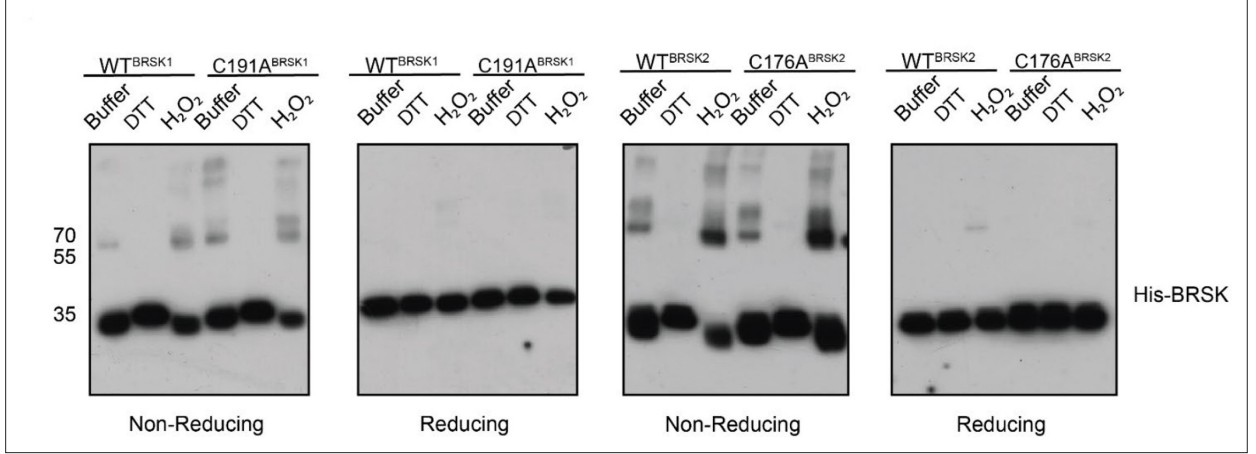

**Figure 7.** BRSK1/2 form limited disulfide-mediated multimers. Western blot analysis of BRSK1/2 kinase domain purified from *E. coli* and incubated with buffer, H₂O₂, or DTT and subjected to non-reducing or reducing PAGE to evaluate the formation of intramolecular disulfide bonds.

The online version of this article includes the following source data and figure supplement(s) for figure 7:

**Source data 1.** Original western blots for *Figure 7*, indicating the relevant bands and treatments.

**Source data 2.** Original files for western blot analysis displayed in *Figure 7*.

**Figure supplement 1.** Evidence for limited BRKS dimer species.

**Figure supplement 1—source data 1.** Original western blots for *Figure 7—figure supplement 1*, indicating the relevant bands and treatments.

**Figure supplement 1—source data 2.** Original files for western blot analysis displayed in *Figure 7—figure supplement 1*.

the G-loop and β3-αC loop, which are distal from the disulfide bonds (*Figure 6—figure supplement 1*).

## Recombinant BRSK proteins form limited protein dimers

Several ARK family members form disulfide bond-dependent dimers (*Nayak et al., 2006*; *Marx et al., 2010*; *Cao et al., 2013*). To evaluate the formation of intermolecular disulfides, we subjected purified kinase domains of BRSK1 and 2 isolated from *E. coli* to non-reducing SDS-PAGE, followed by western blotting to probe for higher order BRSK structures (*Figure 7*). This revealed multiple species of each kinases possessing drastically decreased electrophoretic mobility compared to the major BRSK1/2 monomer bands. These species increased in abundance in the presence of $H_2O_2$ and were absent when DTT was included. Of particular significance was the appearance of a prominent oxidation-dependent species at ~70 kDa, the approximate molecular weight of a BRSK dimer. Consistently, all higher molecular weight species resolved into a single monomer band after reducing (+DTT) SDS-PAGE, which strongly implicates disulfide bond-dependent oligomerization. Curiously, mutation of the T +2 Cys had no discernible effect on the formation of BRSK oligomers, although this is consistent with our previous observation of multiple reactive Cys residues in BRSKs that may be capable of forming a broad variety of intermolecular disulfide bonds. However, it is noteworthy that even in the presence of peroxide, the majority of the BRSK1 and 2 proteins existed as a monomeric species, which suggests that oligomerization is unlikely to be the primary driver of oxidative inhibition that we detect in kinase-based peptide assays. Furthermore, we detected interactions between BRSK1 and BRSK2 (suggesting homo- and heterodimer formation) after co-immunoprecipitation of alternatively tagged variants of the full-length proteins overexpressed in the HEK-293T system (*Figure 7—figure supplement 1a*).

Using SEC-MALS, we confirmed that BRSK1 and 2 (purified in the absence of DTT) were near-uniformly monomeric in solution, but possessed the potential to self-associate and form dimers. The molar mass points across the monomer peak indicates a high degree of homogeneity (weight-average molar mass $M_w$ = ~42 kDa±0.99% and ~43 kDa ± 0.25%, respectively; *Figure 7—figure supplement 1b and c*). Interestingly, the BRSK2 spectra included a high molecular weight shoulder of an approximate dimer size ($M_w$ = ~75 kDa ± 2.1%) that exhibited non-uniform molar mass points indicative of a heterogenous population (likely as a consequence of poor separation between the two peaks and higher order oligomers). SEC-MALS also confirmed the presence of a BRSK species of approximate dimer size ($M_w$ = ~80 kDa) for C183[BRSK2] that was noticeably absent for C176[BRSK2] (*Figure 7—figure supplement 1d and e*). Although the major species observed for both mutants was a monomer, the inability to detect dimer-like peaks for C176[BRSK2] may suggest that the T-+2 Cys plays a role in

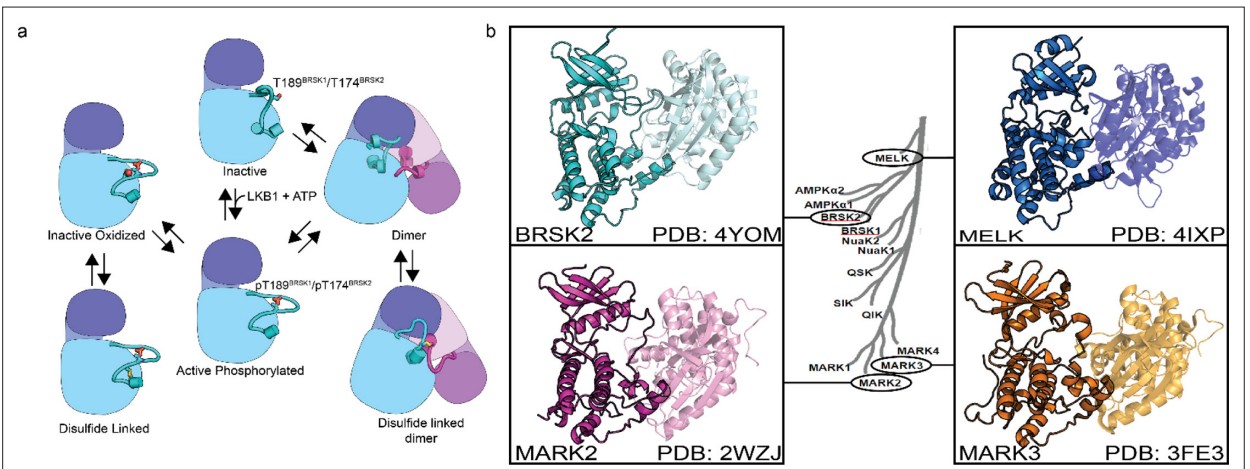

**Figure 8.** Model of BRSK1/2 regulation. (**a**) Schematic diagram demonstrating ways in which residues within BRSK kinases permit fine-tuning of catalytic activity through a variety of oxidative modifications, potentially including inter and intramolecular disulfide bonds. Cartoon representation of kinase domain with N-lobe colored dark blue/purple and the C-lobe colored light blue/purple. (**b**) ARK family member BRSK2, MELK, and MARK2/3 crystal structures demonstrate the ability to form asymmetric dimers, bringing T +2 cys into proximity. Crystal structures for MARK2, and MELK both contain intermolecular disulfide bonds between T +2 cys (*Marx et al., 2010*; *Marx et al., 2006*; *Murphy et al., 2007*; *Cao et al., 2013*).

dimerization. Although we have searched for BRSK1/2 intermolecular disulfide bonds in our LC-MS/MS data in an attempt to characterize the mechanism of dimer formation, we were unable to identify any intermolecular linked peptides. This is likely due to the extremely low abundance of these dimeric species in this sample (thus yielding a very small proportion of inter-linked tryptic peptides) and/or because intermolecular disulfide linked tryptic peptides are too large for identification using this analytical pipeline. Collectively these findings confirm that the isolated kinase domains of both BRSKs primarily occupy a largely monomeric conformation and can form limited higher order redox-sensitive oligomers via covalent S-S bonds in vitro. However, although reversible oxidation-based inactivation of BRSK1 and 2 is apparent in full-length BRSK1 and 2, it remains to be determined to what extent multimerization modulates BRSK catalytic activity (*Figure 8a and b*) or how these mechanisms might contribute to signaling-based interactions in cells.

## Discussion

Redox regulation of kinases and other signaling molecules is a rapidly expanding field of research, which has recently extended far beyond the early observations of oxidative inhibition in protein tyrosine phosphatases (*Brandes et al., 2009*). More recent enquiries have provided strong evidence for direct regulative oxidative modification of Met and Cys residues across divergent protein kinase families, providing temporal and spatial control of their catalytic outputs (*Corcoran and Cotter, 2013*; *Truong and Carroll, 2013*; *Jarvis et al., 2012*). However, despite the prevalence of this regulatory mechanism, the structural basis to explain how redox-active cysteines contribute to allosteric control of catalytic activity is largely unknown. In this study, we demonstrate, for the first time, that two T-loop +2 Cys-containing members of the ARK family, BRSK1 and 2, are reversibly inactivated by oxidative-dependent mechanisms in vitro and in human cells. Moreover, we uncover a multifaceted redox-activity profile for human BRSKs, involving functional Cys-pairs that are conserved within the catalytic domains of these understudied enzymes. In contrast to Ser/Thr kinases such as Aurora A, where a single Cys residue is the dominant driver of redox sensitivity (*Byrne et al., 2020*; *Tsuchiya et al., 2020*), BRSK1 and 2 possess multiple sulfenylation-prone Cys residues. Additionally, the close proximity of distinct Cys 'pairs' permits the formation of two intramolecular disulfide bonds: the first forming between two HRD-motif proximal sites, and the second bridging the conserved T-loop +2 and unique 'CPE' motif Cys residues. We propose a model where disulfide bond formation can impose a steric block on kinase activity while structural perturbations, likely emanating from sulfenylation of conserved BRSK family Cys residues within critical kinase regulatory motifs, provide an additional layer of tunable regulation Validation of these reversibly oxidized Cys species is also of relevance as this may implicate a mechanistic role for ROS sensing in the largely obscure BRSK signaling pathways that operate in different cell types, including those that impact on canonical redox pathways that lead to NRF2 inactivation in cells (*Tamir et al., 2020a*).

### Multilayered redox regulation of BRSKs

The strategic positioning of Cys residues near key regulatory elements in the T-lop suggests an evolutionary adaptation for ROS-based sensing in protein kinases (*Beenstock et al., 2016*; *Pearce et al., 2010*). Full kinase activity typically requires T-loop phosphorylation, a process further modulated by mechanisms like allosteric activation in Aurora A (*Eyers et al., 2003*; *Bayliss et al., 2003*) and activation of CAMKs by CaM (*Rellos et al., 2010*). ARK family kinases, such as BRSK1 and 2, are primed by phosphorylation in the T-loop of a single Thr residue by the master regulatory kinase LKB1. However, our findings suggest that oxidation (or reduction) of key reactive Cys residues in the kinase domains of BRSK1/2 might provide a 'dominant' regulatory oversight of enzyme output whose function in cells is likely controlled by subcellular compartmentalization and/or partner protein interactions.

The ARK family of protein kinases (typified by AMPKα) share a conserved structure with a notable conserved T-loop +2 Cys residue, which is crucial for redox regulation, as evidenced in AMPK and other ARKs (*Byrne et al., 2020*). Mutational analysis of the T +2 Cys position in BRSKs confirms its critical role in supporting kinase activity. Interestingly, a second Cys residue (HRD –7), co-conserved in most ARKs with the exception of MELK, appears to add another regulatory dimension, with alanine substitution at this site significantly reducing BRSK1 and 2 activity (*Figure 4*). This is paralleled by the redox-sensitive Cys 130 in AMPK (*Shao et al., 2014*). Moreover, the tendency of several ARKs to form

disulfide bond-dependent dimers in solution (*Nayak et al., 2006*; *Marx et al., 2010*; *Cao et al., 2013*) is corroborated by crystal structures of MELK, MARK2, MARK3, and BRSK2 (*Figure 8b*), revealing asymmetric dimers linked by disulfide bridges at the T-loop +2 Cys (*Marx et al., 2010*; *Marx et al., 2006*; *Murphy et al., 2007*; *Cao et al., 2013*).

## BRSK-specific adaptations relevant to Cys-based signaling?

BRSKs are differentiated from other ARKs and ePKs through augmentation with two unique Cys residues at the HRD −1 and T-loop +9 positions, forming a novel distinguishing 'CPE' motif in place of the typical APE motif found in most human protein kinases. These Cys pairs facilitate intramolecular disulfide bond formation, influencing kinase activity and conformation. This mechanism is reminiscent of AKT and MELK, where intramolecular disulfide bonds regulate kinase activity (*Murata et al., 2003*; *Huang et al., 2003*; *Byrne et al., 2020*; *Beullens et al., 2005*). Deletion of the T-loop Cys in the CPE motif partially reduces BRSK1 and 2 auto-inhibition, suggesting multiple functional roles for these cysteines, which is supported by our MD simulations.

Comparative evolutionary analysis indicates that approximately 1.4% of all ePKs, including AKT and MELK, have cysteines at the DFG +2 and T-loop +2 positions capable of forming similar disulfide bridges (*Byrne et al., 2020*; *Cao et al., 2013*; *Huang et al., 2003*). MELK, despite lacking the typical HRD −7 Cys of ARKs, has compensatory activation loop Cys residues that form variable disulfide bonds (*Beullens et al., 2005*). A broader analysis across human protein kinases shows proximal Cys pairs capable of forming disulfide bonds in 138 distinct kinases, suggesting a widespread regulatory mechanism in the kinome (*Supplementary file 1*). Our findings in BRSK1 and 2 highlight an extensive intramolecular disulfide network, serving as a reversible switch for kinase activity and interaction regulation. When considering the dominant regulative role of the T-loop available T +2 Cys, it is tempting to speculate that the formation of intramolecular disulfides bonds with adjacent cysteines may be protective against deleterious hyper-oxidized species and enable rapid reactivation of the kinase after emergence from oxidative stress conditions by the disulfide reductase system (*Krishnan et al., 2011*; *Barrett et al., 1999*; *Chen et al., 2008*).

The identification and characterization of unique reactive Cys residues within the kinase domains of BRSK1 and 2 reveal sites of covalent-oxidative modification that may also provide an underexploited opportunity to develop targeted therapeutic strategies for BRSK-associated pathologies. Furthermore, mapping the spatial distribution of Cys across the AMPK-related kinase family provides valuable insights into potential redox hotspots that may underpin a tunable modulation of catalytic outputs with wider implications for cellular signaling. As a master regulator of metabolic homeostasis, AMPK activity is central to appropriate redox balance within cells (*Ren and Shen, 2019*; *Choi et al., 2001*; *Hawley et al., 2010*; *Zmijewski et al., 2010*; *Hinchy et al., 2018*; *Auciello et al., 2014*; *Shao et al., 2014*), but until recently evidence of crosstalk between BRSKs and redox signaling has been less clear. However, BRSKs can indirectly modulate the cellular antioxidant response by orchestrating suppression of the transcription factor (and master regulator/sensor of the antioxidant response), NRF2, in an mTOR-dependent manner in HEK-293 cells (*Tamir et al., 2020a*). NRF2 is targeted for proteasomal degradation by its inhibitor partner, KEAP1, and under conditions of elevated ROS, oxidation of sensor Cys residues in KEAP1 allows NRF2 to escape ubiquitination and induce transcription of the antioxidant machinery (*Baird and Yamamoto, 2020*). Our discovery of redox regulation in BRSKs suggests that it may be part of a multi-protein Cys-based 'relay' network of ROS-sensitive effectors upstream of NRF2, potentially constituting a new oxidative stress signaling mechanism. Uncoupling the specific role of BRSKs in this pathway will be critical in illuminating BRSK1 and 2 physiology and their roles in neuronal function and disease and may simultaneously provide an explanation for the appearance of two functional BRSK1/2 genes in vertebrates.

# Materials and methods
## Recombinant proteins and general reagents

All purchased biochemicals were of the highest purity available, and all recombinant proteins were analyzed by intact mass-spectrometry to confirm the species present. Active, recombinant full-length BRSK1 (2-778) and BRSK2 (2-674) proteins purified from insect Sf21 cells were purchased from MRC PPUU reagents (University of Dundee). Active recombinant LKB1/STRADα/MO25α was purchased

from Merck. Gateway pENTR plasmids encoding full-length human BRSK1 & BRSK2 were generated as part of the NIH common fund initiative to Illuminate the Druggable Genome (IDG) and was a gift from Dr. Ben Major (Washington University, St. Louis). Antibodies for BRSK1 (#5935), BRSK2 (#5460), DYKDDDDK Tag (D6WB5, #14793), phospho-AMPKα (Thr172) (#2535), HA-Tag (C29F4, #3724), 6XHiS tag (#2365), and GAPDH (#2118) were from Cell Signaling Technology (#2365). Antibodies for Phospho-Tau (rabbit, 44-750G) and GFP (mouse, MA5-15256) were from Invitrogen. The glutathione (ab9443) antibody was obtained from Abcam.

## Cloning, gateway recombination, and site-directed mutagenesis

BRSK1 and 2 were cloned into pDest vectors (to express N-terminal Flag or HA-tagged proteins) using the Gateway LR Clonase II system (Invitrogen) as per the manufacturer's instructions. pENtR clones were obtained in the form of a Gateway-compatible donor vectors from the laboratory of Ben Major (Washington University in St. Louis). The Gateway LR Clonase II enzyme mixture mediates recombination between the attL sites on the Entry clone and the attR sites on the destination vector. BRSKs were also cloned into a pcDNA3 vector using a standard T4-ligase (NEB) protocol and expressed in frame with a 3C-protease cleavable N-terminal tandem STREP-tag. The catalytic domains of BRSK1$^{29-358}$ or BRSK2$^{14-341}$ were sub-cloned into pET28a (Novagen) to generate N-terminal hexa-His tagged plasmid constructs for expression of BRSK1/2 catalytic domains in *E. coli*. Site-directed mutagenesis was performed using standard PCR-based mutagenic procedures with the Q5 Site-Directed Mutagenesis Kit (New England Biolabs) following the manufacturer's instructions. All plasmids were validated by complete sequencing of the protein coding region.

## Recombinant BRSK expression and purification

Recombinant human BRSK1$^{29-358}$ or BRSK2$^{14-341}$ proteins, or each of the indicated amino acid substitutions, were produced in BL21 (DE3) pLysS *E. coli* cells (Novagen) and purified in the absence of reducing agents, unless stated otherwise. BRSK1/2 expression was induced with 0.5 mM isopropyl-β-D-thiogalactopyranoside for 18 h at 18°C and N-terminal His6-tag fusion proteins purified by step-wise affinity chromatography and size-exclusion chromatography using a HiLoad 16/600 Superdex 200 column (GE Healthcare) equilibrated in 50 mM Tris-HCl (pH 7.4), 100 mM NaCl, and 10% (v/v) glycerol. Where appropriate for redox assays, recombinant proteins were purified under reducing conditions in the presence of 1 mM DTT, as previously described (*Byrne et al., 2020*). BRSK proteins expressed from bacteria are unphosphorylated and catalytically inactive, and were activated by incubation with 10 ng of purified LKB1/STRADα/MO25α holoenzyme complex in the presence of 1 mM ATP and 10 mM MgCl$_2$ for 18 h at 4°C. Phosphorylation of BRSK proteins was verified by mass spectrometry and/or western blotting analysis using a pThr$^{172}$ AMPKα antibody, which demonstrates cross-reactivity for BRSK1/2 T-loop phosphorylation (*Tamir et al., 2020a*).

## Detection of glutathionylated proteins by immunoblotting

Recombinant BRSK1 and 2 (0.5 µg) were incubated with 50 mM Tris-HCl (pH 7.4) and 100 mM NaCl, with 10 mM GSSG or GSH for 30 min at 20°C, and glutathione-protein complexes were detected by immunoblotting after nonreducing SDS-PAGE.

## BRSK1/2 kinase assays

BRSK activity assays were performed using microfluidic real-time mobility shift-based assays, as described previously (*Byrne et al., 2020*; *Byrne et al., 2016*; *Mohanty et al., 2016*), in the presence of 2 µM of the fluorescent-tagged BRSK1/2 peptide substrate (AMARA; 5-FAM- AMARAASAAA-LARRR -COOH) and 1 mM ATP. Optimal pressure and voltage settings were established to improve separation of phosphorylated and nonphosphorylated peptides. All assays were performed in 50 mM HEPES (pH 7.4), 0.015% (v/v) Brij-35, and 5 mM MgCl$_2$, and the real-time or end point degree of peptide phosphorylation was calculated by differentiating the ratio of the phosphopeptide:peptide. BRSK1/2 activity in the presence of different redox reagents was quantified by monitoring the generation of phosphopeptide during the assay, relative to controls. Data were normalized with respect to control assays, with phosphate incorporation into the peptide generally limited to <20% to prevent depletion of ATP and to ensure assay linearity. Recovery of BRSK activity from oxidative inhibition was assessed by incubating BRSKs with 1 mM hydrogen peroxide, followed by the infusion of 2 mM DTT

and substrate phosphorylation monitoring in real time. To account for potential variability in LKB1-dependent phosphorylation of BRSK proteins, rates of kinase activity (calculated as pmol phosphate incorporation per min) for each protein was normalized by densitometry to the activation site of phosphorylation signal (established with pThr$^{172}$ AMPKα antibodies and ImageJ software).

## Differential scanning fluorimetry

Thermal shift assays were performed with a StepOnePlus real-time polymerase chain reaction (PCR) machine (Life Technologies) using SYPRO Orange dye (Invitrogen) and thermal ramping (0.3°C in step intervals between 25° and 94°C). All proteins were diluted to a final concentration of 5 µM in 50 mM Tris-HCl (pH 7.4) and 100 mM NaCl in the presence or absence of 10 mM DTT and were assayed as described previously (*Foulkes et al., 2018*). Normalized data were processed using the Boltzmann equation to generate sigmoidal denaturation curves, and average $T_m/\Delta T_m$ values were calculated as previously described (*Murphy et al., 2014*) using GraphPad Prism software.

## Human cell culture and treatment

HEK-293T cells were cultured in Dulbecco's modified Eagle medium (Lonza) supplemented with 10% fetal bovine serum (HyClone), penicillin (50 U/ml), and streptomycin (0.25 µg/ml) (Lonza) and maintained at 37°C in 5% $CO_2$ humidified atmosphere. To examine the effects of oxidative stress on BRSK activity, cells were transiently co-transfected for 24 h with plasmids for expression of full-length, N-terminal tagged (Flag, HA or tandem Strep tag) BRSK1/2 (or Cys-Ala mutants) and EGFP-TAU (Addgene), using 3:1 polyethylenimine (average $M_w$, ~25,000 Da; Sigma-Aldrich) to total DNA ratio (4 µg BRSK and 2 µg TAU DNA) in a single well of a 24-well culture plate. To investigate the inactivation of BRSK by peroxide, cells were incubated for 20 min with 10 mM $H_2O_2$, or buffer control. To establish reversibility of oxidative inhibition, cells were incubated for 20 min with 10 mM $H_2O_2$, or buffer control followed by a 15 min incubation with 20 mM reduced glutathione (GSH). In all assays, cells were subsequently washed 3× in PBS, harvested in bromophenol blue-free SDS sample buffer supplemented with 1% Triton X-100, protease inhibitor cocktail tablet, and a phosphatase inhibitor tablet (Roche), or in lysis buffer (50 mM Tris-HCl [pH 7.4], 150 mM NaCl, 1 mM EDTA with 10% [v/v] glycerol and 1% [v/v] Triton X-100, with 1X protease inhibitor cocktail and 1X HALT phosphatase inhibitor). Lysates were sonicated briefly and clarified by centrifugation at 20,817 × *g* for 20 min at 4°C, and supernatants were sampled and diluted 30-fold for calculation of the protein concentration using the Coomassie Plus Staining Reagent (Bradford) Assay Kit (Thermo Fisher Scientific). Cell lysates were normalized for total protein concentration and processed for immunoblotting or immuno-precipitation (IP).

## Liquid chromatography mass spectrometry (LC-MS) analysis BRSKs

48 h post-transfection, HEK-293T cells overexpressing BRSK1 and 2 (containing an N-terminal 3C cleavable tandem STREP-tag) were treated with 1 mM of the cell-permeable chemical oxidant pervanadate for 30 min. Cells were resuspended in ice-cold lysis buffer (50 mM Tris-HCl [pH 6.5], 150 mM NaCl, 10% [v/v] glycerol, 1% [v/v] NP-40, 100 mM iodoacetamide) and disrupted by passing the cell suspension through a 25-gauge needle 10 times. Lysates were clarified by centrifugation at 20,817 × *g* for 20 min at 4°C, and recombinant proteins were affinity precipitated using Strep-TACTIN beads and physically eluted using 3C protease for subsequent MS analysis. Affinity-precipitated BRSK1/2 and bacterially derived recombinant proteins (10 µg) were diluted (~4-fold and ~18-fold respectively) in 100 mM ammonium bicarbonate (pH 8.0) containing 10 mM iodoacetamide and incubated in the dark for 30 min at room temperature. Samples were subjected to an SP3-based Trypsin digestion protocol (adapted from *Daly et al., 2023*) using 100 mM ammonium bicarbonate (pH 8.0) and 0.5 µg of Trypsin gold (Promega). Digested fractions were split 50/50, and one half was treated with dithiothreitol and iodoacetamide as previously described by *Ferries et al., 2017*. Samples were then subjected to in-house packed strong-cation exchange stage tip clean up, as described by *Daly et al., 2021*. Dried peptides were solubilized in 20 µl of 3% (v/v) acetonitrile and 0.1% (v/v) TFA in water, sonicated for 10 min, and centrifuged at 13,000 × *g* for 15 min at 4°C prior to reversed-phase HPLC separation using an Ultimate3000 nano system (Dionex) over a 60 min gradient, as described by *Ferries et al., 2017*. For affinity-precipitated BRSK preparations from human cells, all data acquisition was performed using a Thermo QExactive mass spectrometer (Thermo Scientific), with higher-energy C-trap dissociation (HCD) fragmentation set at 30% normalized collision energy for 2+ to 4+ charge states. MS1 spectra

were acquired in the Orbitrap (70K resolution at 200 *m/z*) over a range of 300–2000 *m/z*, AGC target = 1e6, maximum injection time = 250 ms, with an intensity threshold for fragmentation of 1e3. MS2 spectra were acquired in the Orbitrap (17,500 resolution at 200 *m/z*), maximum injection time = 50 ms, AGC target = 1e5 with a 20 s dynamic exclusion window applied with a 10 ppm tolerance. For bacterially derived recombinant proteins, all data acquisition was performed using a Thermo Fusion Tribrid mass spectrometer (Thermo Scientific), with higher-energy C-trap dissociation (HCD) fragmentation set at 32% normalized collision energy for 2+ to 5+ charge states. MS1 spectra were acquired in the Orbitrap (120K resolution at 200 *m/z*) over a range of 400–2000 *m/z*, AGC target = 100%, maximum injection time = auto, with an intensity threshold for fragmentation of 2.5e4. MS2 spectra were acquired in the Orbitrap (30k resolution at 200 *m/z*), maximum injection time = dynamic, AGC target = auto with a 20 s dynamic exclusion window applied with a 10 ppm tolerance. For disulfide analysis (regardless of sample type), raw data files were converted into mgf format using MSConvert, with peak picking filter set to '2-' and searched with the MASCOT search engine *Perkins et al., 1999*; searching the UniProt Human Reviewed database (updated weekly, accessed January 2023) (*Bateman et al., 2023*) with variable modifications = carbamidomethylation (C), oxidation (M), phosphorylation (ST), instrument type = electrospray ionization–Fourier-transform ion cyclotron resonance (ESI-FTICR) with internal fragments from 200 to 2000 *m/z*, MS1 mass tolerance = 10 ppm, MS2 mass tolerance = 0.01 Da. The crosslinking option was selected for the accessions Q8TDC3 or Q8IWQ3 with strategy set to Brute-force, for InterLink, IntraLink, and LoopLink for the linker 'Xlink: Disulfide (C)'. For the best MASCOT scoring peptide spectrum match for a disulfide-containing peptide, the mgf file was extracted from the raw file and imported into a custom R script for redrawing and manual annotation. Immunoprecipitated samples were additionally analyzed using PEAKS Studio (version XPro) using the same database, mass tolerances, and modifications as previously described. PEAKS-specific search settings: instrument = Orbi-Orbi, Fragmentation = HCD, acquisition = DDA, De Novo details = standard and a maximum of five variable PTMs possible. PEAKS PTM mode was enabled and filtering parameters of De Novo score >15, -log10p(value)>30.0, Ascore > 30.0.

## Phylogenetic analysis

We identified and aligned diverse BRSK-related sequences from the UniProt reference proteomes database (downloaded on June 7, 2022) (*Bateman et al., 2023*) using MAPGAPS (*Neuwald, 2009*). From these hits, we manually curated a diverse set of sequences, then inferred a maximum-likelihood phylogenetic tree with IQ-TREE version 2.0.7 (*Minh et al., 2020*). Branch support values were generated using ultrafast bootstrap (*Hoang et al., 2018*) with 1000 resamples. The optimal substitution model was LG +R6 based on the Bayesian information criterion as determined by ModelFinder (*Kalyaanamoorthy et al., 2017*). The consensus tree was used as our final topology. Subsequent analyses were performed using the ETE3 Toolkit (*Huerta-Cepas et al., 2016*).

## Molecular dynamics simulations

The starting model for MD simulations was selected to provide an accurate representation of the protein kinase in its active-like conformation. To achieve this, we utilized an AlphaFold model of the BRSK2 kinase domain, corresponding to residues 14–267, in an active-like conformation. The average pLDDT score for the portion of the AlphaFold model employed in MD simulations was calculated to be 89.18%, indicating high confidence and accuracy (*Jumper et al., 2021*). Starting structures were prepared using the CHARMM-GUI interface which allowed for incorporation and parameterization of T-loop phosphorylation, cysteine to alanine mutation, and oxidative cysteine modification (*Brooks et al., 2009*; *Lee et al., 2016*; *Jo et al., 2014*). Cysteine 176 (T +2) and 183 (CPE motif) were each mutated to alanine, sulfenic acid, or sulfonic acid forms. The protein was solvated in a cubic box of TIP3P water molecules, and counterions were added to maintain neutrality. The final systems contained ~54,000 atoms.

Prior to production runs, the system was subjected to minimization and equilibration protocols using previously described parameters (*Yeung et al., 2021*; *Venkat et al., 2023*). Initially, a steepest descent energy minimization was performed to relax the system, followed by equilibration at constant volume and temperature (NVT) and constant pressure and temperature (NPT). Each equilibration stage was carried out for 125 ps with 1 fs time steps. Following equilibration, long-range electrostatics were calculated via particle mesh Ewald algorithms using the GROMACS MD engine (*Van Der Spoel*

*et al., 2005*). Three 100 ns production MD replicates were conducted at a 2 fs time step using the CHARMM36 forcefield for each starting model (*Brooks et al., 2009*). The resultant MDs were visualized with PyMOL (*Schrodinger, 2015*) and analyzed in the Python environment (*Michaud-Agrawal et al., 2011*).

## SDS-PAGE and western blotting

Processed cell lysates and purified recombinant proteins were loaded onto 10% (v/v) SDS-PAGE gels, separated by electrophoresis, and transferred onto nitrocellulose membranes using a semi-dry transfer system at 300 mA for 45 min. Nitrocellulose membranes were blocked with 4% (w/v) bovine serum albumin (BSA, Rockland) in Tris-buffered saline with 0.1% (v/v) Tween-20 (TBST) for 1 h at room temperature and incubated overnight at 4°C with the indicated primary antibodies. Protein was detected using specific secondary IRdye conjugated antibodies (donkey anti-rabbit IRdye800cw or goat anti-mouse IRdye680) and imaged using LI-COR Odyssey imaging system, or HRP-conjugated secondary antibodies and enhanced chemiluminescence reagent (Pierce ECL Plus, Thermo Fisher Scientific). All antibodies were prepared in a solution of BSA dissolved in TBST and diluted according to the manufacturer's instructions. Reducing and non-reducing SDS-PAGE for BRSK proteins was performed as previously described (*Byrne et al., 2020*).

Two-color western blot detection method employing infrared fluorescence was used to measure the ratio of Tau phospho serine 262 to total Tau. Total EGFP Tau was detected using a mouse anti GFP antibody and visualized at 680 nm using goat anti-mouse IRdye 680 while phospho-tau was detected using a Tau phospho serine 262-specific antibody and visualized at 800 nm using goat anti rabbit IRdye 800. Imaging was performed using a LI-COR Odyssey Clx with scan control settings set to 169 µm, medium quality, and 0.0 mm distance. Quantification was performed using Li-COR image studio on the raw image files. Total Tau to phospho Tau ratio was determined by measuring the ratio of the fluorescence intensities at 800 nm (pTau) to those at 680 nm (total tau) for each band. Statistical analysis was conducted in GraphPad Prism to determine significant differences between experimental groups. Data is presented as mean ± SEM.

## Size-exclusion chromatography with multi-angle light scattering (SEC-MALS)

The oligomeric state of recombinant BRSKs was characterized by in-line size-exclusion chromatography-multi-angle laser light scattering (SEC-MALS). Purified BRSK proteins (1 mg/ml) were applied directly to a HiLoad 16/60 Superdex 200 attached to an ÄKTA pure fast protein liquid chromatography (FPLC) system equilibrated in 10 mM Tris-HCl pH 7.4, 150 mM NaCl at a flow rate of 0.7 ml/min. Eluted protein was detected by a MALLS detector and a differential refractive index detector (DAWN HELEOS-II and Optilab TrEX; Wyatt Technology, Santa Barbara, CA). Data was analyzed using ASTRA v6.1 software (WYATT). The system was calibrated using BSA prior to data collection with BRSK1/2 proteins.

## Acknowledgements

Funding from NK (grant no: R35 GM139656) is acknowledged. PAE acknowledges funding from the University of Liverpool BBSRC MRC/IAA awards. AV acknowledges funding from the ARCS Foundation. DPB, LAD, SOO, PAE, and CEE also acknowledge BBSRC grants BB/S018514/1, BB/N021703/1, BB/X002780/1, and North West Cancer Research (NWCR) grant CR1208. The content is solely the responsibility of the authors and does not necessarily represent the official views of the National Institutes of Health.

## Additional information

### Funding

| Funder | Grant reference number | Author |
| --- | --- | --- |
| National Institutes of Health | R35 GM139656 | Natarajan Kannan |

| Funder | Grant reference number | Author |
|---|---|---|
| University of Liverpool | BBSRC MRC/IAA awards | Patrick A Eyers |
| ARCS Foundation | | Aarya Venkat |
| Biotechnology and Biological Sciences Research Council | BB/S018514/1 | Dominic P Byrne<br>Leonard A Daly<br>Sally O Oswald<br>Patrick A Eyers<br>Claire E Eyers |
| Biotechnology and Biological Sciences Research Council | BB/N021703/1 | Dominic P Byrne<br>Leonard A Daly<br>Sally O Oswald<br>Patrick A Eyers<br>Claire E Eyers |
| Biotechnology and Biological Sciences Research Council | BB/X002780/1 | Dominic P Byrne<br>Leonard A Daly<br>Sally O Oswald<br>Patrick A Eyers<br>Claire E Eyers |
| North West Cancer Research | CR1208 | Patrick A Eyers<br>Claire E Eyers |

The funders had no role in study design, data collection and interpretation, or the decision to submit the work for publication.

## Author contributions

George N Bendzunas, Data curation, Formal analysis, Investigation, Methodology, Writing – original draft, Writing – review and editing; Dominic P Byrne, Formal analysis, Validation, Investigation, Visualization, Methodology, Writing – original draft, Writing – review and editing; Safal Shrestha, Sally O Oswald, Resources, Software, Methodology; Leonard A Daly, Resources, Software, Formal analysis, Methodology; Samiksha Katiyar, Resources, Methodology; Aarya Venkat, Data curation, Software, Methodology; Wayland Yeung, Software, Visualization, Methodology; Claire E Eyers, Resources, Formal analysis, Methodology; Patrick A Eyers, Formal analysis, Supervision, Funding acquisition, Investigation, Methodology, Project administration, Writing – review and editing; Natarajan Kannan, Conceptualization, Formal analysis, Supervision, Funding acquisition, Project administration, Writing – review and editing

## Author ORCIDs

George N Bendzunas ![orcid] https://orcid.org/0000-0001-5986-7862
Dominic P Byrne ![orcid] http://orcid.org/0000-0001-5197-345X
Aarya Venkat ![orcid] https://orcid.org/0000-0002-8793-4097
Claire E Eyers ![orcid] https://orcid.org/0000-0002-3223-5926
Patrick A Eyers ![orcid] http://orcid.org/0000-0002-9220-2966
Natarajan Kannan ![orcid] https://orcid.org/0000-0002-2833-8375

Reviewer #1 (Public Review): https://doi.org/10.7554/eLife.92536.4.sa1
Reviewer #2 (Public Review): https://doi.org/10.7554/eLife.92536.4.sa2
Author response https://doi.org/10.7554/eLife.92536.4.sa3

# Additional files

## Supplementary files

Supplementary file 1. Proximal cysteine pairs in the human kinome predicted from structural analysis of human protein kinases in the AlphaFold database. Cysteine pairs within 10 Å are considered proximal.

MDAR checklist

## Data availability

All data generated in this study are included in the manuscript. All mass spectrometry data has been deposited at the ProteomeXchange Consortium (http://proteomecentral.proteomexchange.org) via the PRIDE partner repository with the dataset identifiers PXD044990. Source data are provided for each figure, and molecular dynamics simulation trajectories are made available at https://doi.org/10.5061/dryad.gb5mkkx1j.

The following datasets were generated:

| Author(s) | Year | Dataset title | Dataset URL | Database and Identifier |
|---|---|---|---|---|
| Bendzunas G | 2025 | Redox regulation of brain selective kinases BRSK1/2: Implications for dynamic control of the eukaryotic AMPK family through Cys-based mechanisms | https://doi.org/10.5061/dryad.gb5mkkx1j | Dryad Digital Repository, 10.5061/dryad.gb5mkkx1j |
| Daly L, Eyers C | 2025 | Redox Regulation of Brain Selective Kinases BRSK1/2: Implications for Dynamic Control of the Eukaryotic AMPK family through Cys-based mechanisms | https://www.ebi.ac.uk/pride/archive/projects/PXD044990 | PRIDE, PXD044990 |

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
