## [Editor Report · eLife assessment]

This study provides **fundamental** new knowledge into the role of reversible cysteine oxidation and reduction in protein kinase regulation. The data provide **convincing** evidence that intramolecular disulfide bonds serve a repressive regulatory role in the brain-selective kinases (BRSK) 1 and 2; part of the as-yet understudied ‘dark kinome’. The findings will be of broad interest to biochemists, structural biologists, and those interested in the rational design and development of next-generation kinase inhibitors.

---

## [Referee Report · Reviewer #1 (Public Review)]

Summary:

Bendzunas, Byrne et al. explore two highly topical areas of protein kinase regulation in this manuscript. Firstly, the idea that Cys modification could regulate kinase activity. The senior authors have published some standout papers exploring this idea of late, and the current work adds to the picture of how active site Cys might have been favoured in evolution to serve critical regulatory functions. Second, BRSK1/2 are understudied kinases listed as part of the "dark kinome" so any knowledge of their underlying regulation is of critical importance to advancing the field.

Strengths:

In this study, the author pinpoints highly-conserved, but BRSK-specific, Cys residues as key players in kinase regulation. There is a delicate balance between equating what happens in vitro with recombinant proteins relative to what the functional consequence of Cys mutation might be in cells or organisms, but the authors are very clear with the caveats relating to these connections in their descriptions and discussion. Accordingly, by extension, they present a very sound biochemical case for how Cys modification might influence kinase activity in cellular environs.

Comments on revised version:

The authors have satisfactorily addressed my concerns.

---

## [Referee Report · Reviewer #2 (Public Review)]

Summary:

In this study by Bendzunas et al, the authors show that the formation of intra-molecular disulfide bonds involving a pair of Cys residues near the catalytic HRD motif and a highly conserved T-Loop Cys with a BRSK-specific Cys at an unusual CPE motif at the end of the activation segment function as repressive regulatory mechanisms in BSK1 and 2. They observed that mutation of the CPE-Cys only, contrary to the double mutation of the pair, increases catalytic activity in vitro and drives phosphorylation of the BRSK substrate Tau in cells. Molecular modeling and molecular dynamics simulations indicate that oxidation of the CPE-Cys destabilizes a conserved salt bridge network critical for allosteric activation. The occurrence of spatially proximal Cys amino acids in diverse Ser/Thr protein kinase families suggests that disulfide-mediated control of catalytic activity may be a prevalent mechanism for regulation within the broader AMPK family. Understanding the molecular mechanisms underlying kinase regulation by redox-active Cys residues is fundamental as it appears to be widespread in signaling proteins and provides new opportunities to develop specific covalent compounds for the targeted modulation of protein kinases.

The authors demonstrate that intramolecular cysteine disulfide bonding between conserved cysteines can function as a repressing mechanism as indicated by the effect of DTT and the consequent increase in activity by BSK-1 and -2 (WT). The cause-effect relationship of why mutation of the CPE-Cys only increases catalytic activity in vitro and drives phosphorylation of the BRSK substrate Tau in cells is not clear to me. The explanation given by the authors based on molecular modeling and molecular dynamics simulations is that oxidation of the CPE-Cys (that will favor disulfide bonding) destabilizes a conserved salt bridge network critical for allosteric activation. However, no functional evidence of the impact of the salt-bridge network is provided. If you mutated the two main Cys-pairs (aE-CHRD and A-loop T+2-CPE) you lose the effect of DTT, as the disulfide pairs cannot be formed, hence no repression mechanisms take place, however when looking at individual residues I do not understand why mutating the CPE only results in the opposite effect unless it is independent of its connection with the T+2residue on the A-loop.

Strengths:

This is an important and interesting study providing new knowledge in the protein kinase field with important therapeutic implications for the rationale design and development of next-generation inhibitors.

Comments on revised version:

The authors have satisfactorily addressed my concerns.

---

## [Author Response]

The following is the authors’ response to the previous reviews.

**Reviewer #1:**
Comments on revised version:The authors have satisfactorily addressed my concerns.I suggest some minor edits, however. Line 747 does not mention MARK3 and neither does the figure 8 legend (just MARK2). It would be helpful if the authors could include references to the papers reporting the shown structures in the Figure 8 legend

We have added MARK3 and related references in the revised Figure 8 legend.

**Reviewer #2:**
I would recommend that the catalog numbers from the different antibodies used in the study, mainly CST and Invitrogen are depicted in material and methods (see Methods/Recombinant proteins and general reagents).

Thank you for the comment. We have now added the antibody catalog numbers in the revised methods section.

I have one remark related to question number 5 (my question was not clear enough). I meant if the authors did look at the functional relevance of the residues implicated in the identified salt-bridge network/tethers. What happens to the proteins functionally when you mutate those residues? (represented on Fig. 8).Otherwise, the authors have satisfactorily addressed my concerns.

Yes, we have analyzed the stability of the salt bridge interaction in the context of cysteine mutations, and our findings are described in the results section titled “Cysteine mutations alter critical structural interactions required for kinase allosteric regulation (Figure 6)”. However, we have not performed mutational analysis of the salt bridge residues as we feel this would be beyond the scope of the current study.